# Disentangling Identifiable Features from Noisy Data with Structured Nonlinear ICA

**Hermanni Hälvä**[1] *    **Sylvain Le Corff**[2]    **Luc Lehéricy**[3]

**Jonathan So**[4]    **Yongjie Zhu**[1]    **Elisabeth Gassiat**[5] †    **Aapo Hyvärinen**[1] †

[1]Department of Computer Science, University of Helsinki, Finland
[2] Samovar, Télécom SudParis, département CITI, Institut Polytechnique de Paris, Palaiseau, France
[3]Laboratoire J. A. Dieudonné, Université Côte d'Azur, CNRS, 06100, Nice, France
[4]Department of Engineering, University of Cambridge, UK
[5]Université Paris-Saclay, CNRS, Laboratoire de mathématiques d'Orsay, 91405, Orsay, France

## Abstract

We introduce a new general identifiable framework for principled disentanglement referred to as Structured Nonlinear Independent Component Analysis (SNICA). Our contribution is to extend the identifiability theory of deep generative models for a very broad class of structured models. While previous works have shown identifiability for specific classes of time-series models, our theorems extend this to more general temporal structures as well as to models with more complex structures such as spatial dependencies. In particular, we establish the major result that identifiability for this framework holds even in the presence of noise of unknown distribution. Finally, as an example of our framework's flexibility, we introduce the first nonlinear ICA model for time-series that combines the following very useful properties: it accounts for both nonstationarity and autocorrelation in a fully unsupervised setting; performs dimensionality reduction; models hidden states; and enables principled estimation and inference by variational maximum-likelihood.

## 1 Introduction

A central tenet of unsupervised deep learning is that noisy and high dimensional real world data is generated by a nonlinear transformation of lower dimensional latent factors. Learning such lower dimensional features is valuable as they may allow us to understand complex scientific observations in terms of much simpler, semantically meaningful, representations (Morioka et al., 2020; Zhou and Wei, 2020). Access to a ground truth generative model and its latent features would also greatly enhance several other downstream tasks such as classification (Klindt et al., 2021; Banville et al., 2021), transfer learning (Khemakhem et al., 2020b), as well as causal inference (Monti et al., 2019; Wu and Fukumizu, 2020).

A recently popular approach to deep representation learning has been to learn *disentangled* features. Whilst not rigorously defined, the general methodology has been to use deep generative models such as VAEs (Kingma and Welling, 2014; Higgins et al., 2017) to estimate semantically distinct factors of variation that generate and encode the data. A substantial problem with the vast majority of work on disentanglement learning is that the models used are not *identifiable* – that is, they do not learn the true generative features, even in the limit of infinite data – in fact, this task has been proven

---

*hermanni.halva@helsinki.fi
†Equal senior authorship

35th Conference on Neural Information Processing Systems (NeurIPS 2021).

impossible without inductive biases on the generative model (Hyvärinen and Pajunen, 1999; Locatello et al., 2019). Lack of identifiability plagues deep learning models broadly and has been implicated as one of the reasons for unexpectedly poor behaviour when these models are deployed in real world applications (D'Amour et al., 2020). Fortunately, in many applications the data have dependency structures, such as temporal dependencies which introduce inductive biases. Recent advances in both identifiability theory and practical algorithms for nonlinear ICA (Hyvärinen and Morioka, 2016, 2017; Hälvä and Hyvärinen, 2020; Morioka et al., 2021; Klindt et al., 2021; Oberhauser and Schell, 2021) exploit this and offer a principled approach to disentanglement for such data. Learning statistically independent nonlinear features in such models is well-defined, i.e. those models are identifiable.

However, the existing nonlinear ICA models suffer from numerous limitations. First, they only exploit specific types of temporal structures, such as either temporal dependencies or nonstationarity. Second, they often work under the assumption that some 'auxiliary' data about a *latent* process is observed, such as knowledge of the switching points of a nonstationary process as in Hyvärinen and Morioka (2016); Khemakhem et al. (2020a) . Furthermore, all the nonlinear ICA models cited above, with the exception of Khemakhem et al. (2020a), assume that the data are fully observed and noise-free, even though observation noise is very common in practice, and even Khemakhem et al. (2020a) assumes the noise distribution to be exactly known. This approach of modelling observation noise explicitly is in stark contrast to the approach taken in papers, such as Locatello et al. (2020), who instead consider general stochasticity of their model to be captured by latent variables – this approach would be ill-suited to the type of denoising one would often need in practice. Lastly, the identifiability theorems in previous nonlinear ICA works usually restrict the latent components to a specific class of models such as exponential families (but see Hyvärinen and Morioka (2017)).

In this paper we introduce a new framework for identifiable disentanglement, Structured Nonlinear ICA (SNICA), which removes each of the aforementioned limitations in a single unifying framework. Furthermore, the framework guarantees identifiability of a rich class of nonlinear ICA models that is able to exploit dependency structures of any arbitrary order and thus, for instance, extends to spatially structured data. This is the first major theoretical contribution of our paper.

The second important theoretical contribution of our paper proves that models within the SNICA framework are identifiable even in the presence of additive output noise of *arbitrary, unknown* distribution. We achieve this by extending the theorems by Gassiat et al. (2020b,a). The subsequent practical implication is that SNICA models can perform dimensionality reduction to identifiable latent components and de-noise observed data. We note that noisy-observation part of the identifiability theory is not even limited to nonlinear ICA but applies to any system observed under noise.

Third, we give mild sufficient conditions, relating to the strength and the non-Gaussian nature of the temporal or spatial dependencies, enabling identifiability of nonlinear independent components in this general framework. An important implication is that our theorems can be used, for example, to develop models for disentangling identifiable features from spatial or spatio-temporal data.

As an example of the flexibility of the SNICA framework, we present a new nonlinear ICA model called $\Delta$-SNICA . It achieves the following very practical properties which have previously been unattainable in the context of nonlinear ICA: the ability to account for both nonstationarity and autocorrelation in a fully unsupervised setting; ability perform dimensionality reduction; model latent states; and to enable principled estimation and inference by variational maximum-likelihood methods. We demonstrate the practical utility of the model in an application to noisy neuroimaging data that is hypothesized to contain meaningful lower dimensional latent components and complex temporal dynamics.

## 2 Background

We start by giving some brief background on Nonlinear ICA and identifiability. Consider a model where the distribution of observed data $\mathbf{x}$ is given by $p_X(\mathbf{x}; \boldsymbol{\theta})$ for some parameter vector $\boldsymbol{\theta}$. This model is called identifiable if the following condition is fulfilled:

$$\forall(\boldsymbol{\theta}, \boldsymbol{\theta}') \qquad p_X(\mathbf{x}; \boldsymbol{\theta}) = p_X(\mathbf{x}; \boldsymbol{\theta}') \Rightarrow \boldsymbol{\theta} = \boldsymbol{\theta}' . \tag{1}$$

In other words, based on the observed data distribution alone, we can *uniquely* infer the parameters that generated the data. For models parameterized with some nonparametric function estimator $\mathbf{f}$, such as a deep neural network, we can replace $\boldsymbol{\theta}$ with $\mathbf{f}$ in the equation above. In practice, identifiability

might hold for some parameters, not all; and parameters might be identifiable up to some more or less trivial indeterminacies, such as scaling.

In a typical nonlinear ICA setting we observe some $\mathbf{x} \in \mathbb{R}^N$ which has been generated by an invertible nonlinear mixing function $\mathbf{f}$ from latent independent components $\mathbf{s} \in \mathbb{R}^N$, with $p(\mathbf{s}) = \prod_{i=1}^{N} p(s^{(i)})$, as per:

$$\mathbf{x} = \mathbf{f}(\mathbf{s}), \tag{2}$$

Identifiability of $\mathbf{f}$ would then mean that we can in theory find the true $\mathbf{f}$, and subsequently the true data generating components. Unfortunately, without some additional structure this model is unidentifiable, as shown by Hyvärinen and Pajunen (1999): there is an infinite number of possible solutions and these have no trivial relation with each other. To solve this problem, previous work (Sprekeler et al., 2014; Hyvärinen and Morioka, 2016, 2017) developed models with temporal structure. Such time series models were generalized and expressed in a succinct way by Hyvärinen et al. (2019); Khemakhem et al. (2020a) by assuming the independent components are *conditionally* independent upon some observed auxiliary variable $u_t$: $p(\mathbf{s}_t|u_t) = \prod_{i=1}^{N} p(s_t^{(i)}|u_t)$. In a time series context, the auxiliary variable might be history, e.g. $u_t = \mathbf{x}_{t-1}$, or the index of a time segment to model nonstationarity (or piece-wise stationarity). (It could also be data from another modality, such as audio data used to condition video data (Arandjelovic and Zisserman, 2017).)

Notice that the mixing function $\mathbf{f}$ in (2) is assumed bijective and thus *identifiable* dimension reduction is not possible in most of the models discussed above. The only exceptions, we are aware of, are Khemakhem et al. (2020a); Klindt et al. (2021) who choose $\mathbf{f}$ as injective rather than bijective. Further, Khemakhem et al. (2020a) assume additive noise on the observations $\mathbf{x} = \mathbf{f}(\mathbf{s}) + \boldsymbol{\varepsilon}$, which allows to estimate posterior of $\mathbf{s}$ by an identifiable VAE (iVAE). We will take a similar strategy in what follows.

## 3 Definition of Structured Nonlinear ICA

In this section, we first present the new framework of Structured Nonlinear ICA (SNICA) – a broad class of models for identifiable disentanglement and learning of independent components when data has structural dependencies. Next, we give an example of a particularly useful specific model that fits within our framework, called $\Delta$-SNICA , by using switching linear dynamical latent processes.

### 3.1 Structured Nonlinear ICA framework

Consider observations $(\mathbf{x}_t)_{t \in \mathbb{T}} = ((x_t^{(1)}, \ldots, x_t^{(M)}))_{t \in \mathbb{T}}$ where $\mathbb{T}$ is a discrete indexing set of arbitrary dimension. For discrete time-series models, like previous works, $\mathbb{T}$ would be a subset of $\mathbb{N}$. Crucially, however, we allow it to be any arbitrary indexing variable that describes a desired structure. For instance, $\mathbb{T}$ could be a subset of $\mathbb{N}^2$ for spatial data.

We assume the data is generated according the following nonlinear ICA model. First, there exist latent components $\mathbf{s}^{(i)} = (s_t^{(i)})_{t \in \mathbb{T}}$ for $i \in \{1, \ldots, N\}$ where for any $t, t' \in \mathbb{T}$, the distributions of $(\mathbf{s}_t^{(i)})_{1 \leqslant i \leqslant N}$ and $(\mathbf{s}_{t'}^{(i)})_{1 \leqslant i \leqslant N}$ are the same, which is a weak form of *stationarity*. Second, we assume that for any $m \in \mathbb{N}^*$ and $(t_1, \ldots, t_m) \in \mathbb{T}^m$, $p(\mathbf{s}_{t_1}, \ldots, \mathbf{s}_{t_m}) = \prod_{i=1}^{N} p(s_{t_1}^{(i)}, \ldots, s_{t_m}^{(i)})$: that is, the components are unconditionally *independent*. We further assume that the nonlinear mixing function $\mathbf{f} : \mathbb{R}^N \to \mathbb{R}^M$ with $M \geqslant N$ is injective, so there may be more observed variables than components. Finally, denote observational noise by $\boldsymbol{\varepsilon}_t \in \mathbb{R}^M$ and assume that they are i.i.d. for all $t \in \mathbb{T}$ and independent of the signals $\mathbf{s}^{(i)}$. Putting these together, we assume the mixing model where for each $t \in \mathbb{T}$,

$$\mathbf{x}_t = \mathbf{f}(\mathbf{s}_t) + \boldsymbol{\varepsilon}_t, \tag{3}$$

where $\mathbf{s}_t = (s_t^{(1)}, \ldots, s_t^{(N)})$. Importantly, $\boldsymbol{\varepsilon}_t$ can have any arbitrary unknown distribution, even with dependent entries; in fact, it may even not have finite moments.

The main appeal of this framework is that, under the conditions given in next section, we can now guarantee identifiability for a very broad and rich class of models.

First, notice that all previous Nonlinear ICA time-series models can be reformulated and often improved upon when viewed through this new unifying framework. In other words, we can create

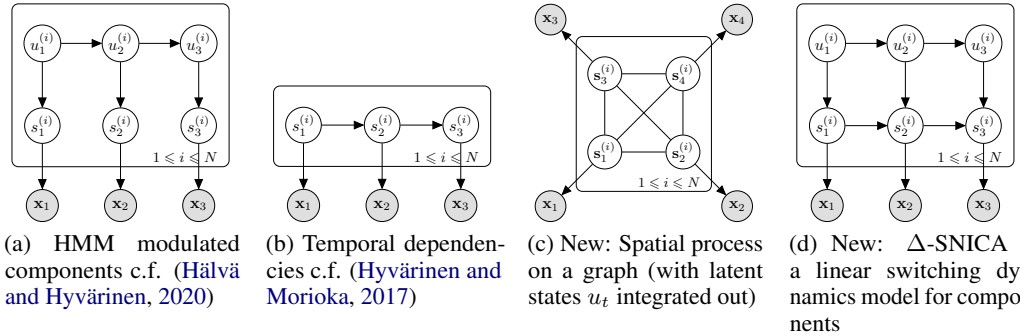

(a) HMM modulated components c.f. (Hälvä and Hyvärinen, 2020)

(b) Temporal dependencies c.f. (Hyvärinen and Morioka, 2017)

(c) New: Spatial process on a graph (with latent states $u_t$ integrated out)

(d) New: $\Delta$-SNICA, a linear switching dynamics model for components

Figure 1: Graphical models for the SNICA framework

models that are very much like those previous works, and capture their dependency profiles, but with the changes that by assuming unconditional independence and output noise we now allow them to perform dimension reduction (this does also require some additional assumptions needed in our identifiability theorems below). To see this, consider the model in Hälvä and Hyvärinen (2020) which captures nonstationarity in the independent components through a global hidden Markov chain. We can transform this model into the SNICA framework if we instead model *each* independent component as its own HMM (Figure 1a), with the added benefit that we now have marginally independent components and are able to perform dimensionality reduction into low dimensional latent components. Nonlinear ICA with time-dependencies, such as in an autoregressive model, proposed by Hyvärinen and Morioka (2017) is also a special case of our framework (Figure 1b), but again with the extension of dimensionality reduction. Furthermore, this framework allows for a plethora of new Nonlinear ICA models to be developed. As described above, these do not have to be limited to time-series but could for instance be a process on a two-dimensional graph with appropriate (in)dependencies (see Figure 1c). However, we now proceed to introduce a particularly useful time-series model using our framework.

### 3.2 $\Delta$-SNICA : Nonlinear ICA with switching linear dynamical systems

While the above framework has great generality, any practical application will need a specific model. Next we propose one which combines the following properties of previous nonlinear ICA models into a single model: ability to account for both nonstationarity and autocorrelation in a fully unsupervised setting, to perform dimensionality reduction and model hidden states. Real world processes, such as video/audio data, financial time-series, and brain signals, exhibit these properties – disentangling latent features in such data would hence be very useful.

Our new model is depicted in Figure 1d. The independent components are generated by a Switching Linear Dynamical System (SLDS) (Ackerson and Fu, 1968; Chang and Athans, 1978; Hamilton, 1990; Ghahramani and Hinton, 2000) with additional latent variables to express rich dynamics. Formally, for each independent component $i \in \{1, \ldots, N\}$, consider the following SLDS over some latent vector $\mathbf{y}_t^{(i)}$:

$$\mathbf{y}_t^{(i)} = \mathbf{B}_{u_t}^{(i)} \mathbf{y}_{t-1}^{(i)} + \mathbf{b}_{u_t}^{(i)} + \boldsymbol{\varepsilon}_{u_t}^{(i)}, \tag{4}$$

where $u_t := u_t^{(i)}$ is a state of a first-order hidden Markov chain $(u_t^{(i)})_{t=1:T}$. Crucially, we assume that the independent components at each time-point are the first elements $y_{t,1}^{(i)}$ of $\mathbf{y}_t^{(i)} = (y_{t,1}^{(i)}, \ldots, y_{t,d}^{(i)})^T$, i.e. $s_t^{(i)} = y_{t,1}^{(i)}$. The rest of the elements in $\mathbf{y}_t^{(i)}$ are latent variables modelling hidden dynamics. The great utility of using such a higher-dimensional latent variable is that this model allows us, for example, as a special case, to consider higher-order ARMA processes, thus modelling each $s_t^{(i)}$ as switching between ARMA processes of an order determined by the dimensionality of $\mathbf{y}_t$. We call the ensuing model $\Delta$-SNICA ("Delta-SNICA", with delta as in "dynamic").

# 4 Identifiability

In this section, we present two very general identifiability theorems for SNICA. We basically decouple the problem into two parts. First, we consider identifying the noise-free distribution of $\mathbf{f}(\mathbf{s}_t)$ from noisy data. Theorem 1 states conditions—on tail behaviour, non-degeneracy, and non-Gaussianity—under which it is possible to recover the distribution of a process based on noisy data with unknown noise distribution. Second, we consider demixing of the nonlinearly mixed data. Theorem 2 provides general conditions—on temporal or spatial dependencies, and non-Gaussianity—that allow recovery of the mixing function $\mathbf{f}$ when there is no more noise. We then consider application of these theorems to SNICA.

## 4.1 Identifiability with unknown noise distribution

Consider the model

$$\mathbf{x}_t = \mathbf{z}_t + \boldsymbol{\varepsilon}_t \,, \tag{5}$$

where $(\mathbf{z}_t)_{t \in \mathbb{T}}$ is a family of random variables in $\mathbb{R}^M$ such that all $\mathbf{z}_t$, $t \in \mathbb{T}$, have the same marginal distribution, and $(\boldsymbol{\varepsilon}_t)_{t \in \mathbb{T}}$ is a family of independent (over $t$) and identically distributed random variables, independent of $(\mathbf{z}_t)_{t \in \mathbb{T}}$. Let $P$ be the common distribution of each $\boldsymbol{\varepsilon}_t$, for $t \in \mathbb{T}$. Let $t_1$ and $t_2$ in $\mathbb{T}$, and consider the following assumptions.

- (A1) [Tail behaviour] For some $\rho < 3$, there exist $A$ and $B$ such that for all $\lambda \in \mathbb{R}^N$,

$$\mathbb{E}[\exp(\langle \lambda, \mathbf{z}_{t_1} \rangle)] \leqslant A \exp(B\|\lambda\|^\rho) \,.$$

- (A2) [Non-degeneracy] For any $\eta \in \mathbb{C}^M$, $\mathbb{E}[\exp\{\langle \eta, \mathbf{z}_{t_2}\rangle\}|\ \mathbf{z}_{t_1}]$ is not the null random variable.

- (A3) [Non-Gaussianity] The following assertion is false: there exist a vector $\eta \in \mathbb{R}^M$ and independent random variables $\tilde{z}$ and $u$, such that $u$ is a non dirac Gaussian random variable and $\langle \eta, \mathbf{z}_{t_1} \rangle$ has the same distribution as $\tilde{z} + u$.

We defer the detailed discussion on the practical meaning of the assumptions (A1-A3) in the context of SNICA to Section 4.3. We next present Theorem 1 which establishes identifiability under unknown noise (its proof is postponed to Section A.1 in the Supplementary Material):

**Theorem 1** *Assume that assumptions (A1), (A2) and (A3) hold for some $(t_1, t_2) \in \mathbb{T}^2$. Then, up to translation, for all $m \geqslant 2$, for all $(t_3, \ldots, t_m) \in \mathbb{T}^{m-2}$, the application that associates the distribution of $(\mathbf{z}_{t_1}, \ldots, \mathbf{z}_{t_m})$ and $P$ to the distribution of $(\mathbf{x}_{t_1}, \ldots, \mathbf{x}_{t_m})$ is one-to-one.*

Here, up to translation means that adding a constant vector to all $\boldsymbol{\varepsilon}_t$, and substracting this constant to all $\mathbf{z}_t$, $t \in \{t_1, \ldots, t_m\}$, does not change the distribution of $(\mathbf{x}_{t_1}, \ldots, \mathbf{x}_{t_m})$. The proof of Theorem 1 extends that of Theorem 1 in (Gassiat et al., 2020b), see also (Gassiat et al., 2020a), which assumed sub-Gaussian noise-free data. Our extension allows the noise-free data to have heavier tails, which is important since (noise-free) data in many real-world applications is super-Gaussian, i.e. heavy-tailed, as is well-known in work on linear ICA (Hyvärinen et al., 2001).

Importantly, there is no assumption on the unknown noise distribution in Theorem 1. In fact, it does not even assume a mixing as in ICA, and thus extends greatly outside of the framework of this paper.

## 4.2 Identifiability of the mixing function

Based on Theorem 1, it is possible to recover the distribution of the noise-free data in SNICA in (3) by setting $\mathbf{z}_t = \mathbf{f}(\mathbf{s}_t)$. Next, we consider under which conditions the mixing function $\mathbf{f}$ is identifiable. Denote by $S = S^{(1)} \times \cdots \times S^{(N)}$ the support of the distribution of all $\mathbf{s}_t$. We consider the situation where each $S^{(i)} \subset \mathbb{R}$, $1 \leqslant i \leqslant N$, is connected, so that each $S^{(i)}$ is an interval. We assume moreover that the injective mixing function $\mathbf{f}$ is a $\mathcal{C}^2$ diffeomorphism between $S$ and a $\mathcal{C}^2$ differentiable manifold $\mathcal{M} \subset \mathbb{R}^M$. Formally, this means that there exists an atlas $\{\varphi_\vartheta : U_\vartheta \to \mathbb{R}^N\}_{\vartheta \in \Theta}$ of $\mathcal{M}$ such that for all $\vartheta, \vartheta' \in \Theta$, the map $\varphi_\vartheta \circ \varphi_{\vartheta'}^{-1}$ is a $\mathcal{C}^2$ map, and $\mathbf{f}$ is a bijection $\mathbb{R}^N \to \mathcal{M}$ such that for all $\vartheta \in \Theta$, $\varphi_\vartheta \circ \mathbf{f}$ and $\mathbf{f}^{-1} \circ \varphi_\vartheta^{-1}$ have continuous second derivatives. The sets $U_\vartheta$, $\vartheta \in \Theta$, cover $\mathcal{M}$ and are open in $\mathcal{M}$. The proof of Theorem 2 is postponed to Section A.2 in the Supplementary Material.

**Theorem 2** *Assume that there exist* $m \geqslant 2$ *and* $(t_1, \ldots, t_m) \in \mathbb{T}^m$ *such that the vector* $(s_{t_1}^{(i)}, \ldots, s_{t_m}^{(i)})$ *has a density* $p_m^{(i)}$ *which is* $\mathcal{C}^2$ *on* $(S^{(i)})^m$. *Assume moreover that there exist* $(k, l) \in \{1, \ldots, m\}^2$ *with* $k \neq l$ *such that the following assumptions hold with* $Q_m^{(i)} = \log p_m^{(i)}$.

- *(B1) (Uniform $(k, l)$-dependency). For all $i \in \{1, \ldots, N\}$, the set of zeros of $\frac{\partial^2}{\partial s_{t_k}^{(i)} \partial s_{t_l}^{(i)}} Q_m^{(i)}$*

  *is a meagre subset of $(S^{(i)})^m$, i.e. it contains no open subset.*

- *(B2) (Local $(k, l)$-non quasi Gaussianity). For any open subset $A \subset S^m$, there exists at most one $i \in \{1, \ldots, N\}$ such that there exists a function $\alpha : \mathbb{R}^{m-1} \to \mathbb{R}$ and a constant $c \in \mathbb{R}$ such that for all $s \in A$,*

$$\frac{\partial^2}{\partial s_{t_k}^{(i)} \partial s_{t_l}^{(i)}} Q_m^{(i)} = c \, \alpha(s_{t_k}^{(i)}, \mathbf{s}_{(-t_k, -t_l)}^{(i)}) \alpha(s_{t_l}^{(i)}, \mathbf{s}_{(-t_k, -t_l)}^{(i)}) \,, \tag{6}$$

  *where $\mathbf{s}_{(-t_k, -t_l)}^{(i)}$ is $(s_{t_1}^{(i)}, \ldots, s_{t_m}^{(i)})$ without the coordinates $t_k$ and $t_l$.*

*Then, $\mathbf{f}^{-1}$ can be recovered up to permutation and coordinate-wise transformations from the distribution of $(\mathbf{f}(\mathbf{s}_{t_1}), \ldots, \mathbf{f}(\mathbf{s}_{t_m}))$.*

## 4.3 Applications to SNICA

In this section, we provide additional comments on the assumptions (A1-A3) and (B1-B2) and their verification in the context of SNICA.

**Assumption (A1)** is a condition on the tails of the noise-free data: it allows tails that are somewhat heavier than Gaussian tails. It is in fact equivalent to assuming that for some $\tilde{\rho} > 3/2$, there exists $A', B' > 0$ such that for all $t > 0$, $\mathbb{P}(\|\mathbf{z}_{t_1}\| \geqslant t) \leqslant A' \exp(-B' t^{\tilde{\rho}})$.

**Assumption (A2)** is a non-degeneracy condition likely to be fulfilled for any randomly chosen SNICA model parameters. As an example, consider a model such as Fig. 1c, where there exist hidden variables $(u_t)_{t \in \mathbb{T}}$ taking values in a finite set $\{1, \ldots, K\}$ such that the pairs of variables $(\mathbf{z}_t, u_t)$ have the same distribution for all $t \in \mathbb{T}$, and such that conditioned on $(u_t)_{t \in \mathbb{T}}$, the variables $(\mathbf{z}_t)_{t \in \mathbb{T}}$ are independent and the distribution of $\mathbf{z}_t$ only depends on $u_t$. (As a special case, this model includes the temporal HMM setting described in Fig. 1a.) Let $(t_1, t_2) \in \mathbb{T}^2$. For all $u, v \in \{1, \ldots, K\}$, let $\pi(u) = p_{u_{t_1}}(u)$ be the mass function of $u_{t_1}$, $Q(u, v) = p_{u_{t_2}|u_{t_1}}(v|u)$ be the transition matrix from $u_{t_1}$ to $u_{t_2}$, and $\gamma_u(\mathbf{z}) = p_{\mathbf{z}_{t_1}|u_{t_1}}(\mathbf{z}|u)$ be the density of $\mathbf{z}_{t_1}$ conditionally to $u_{t_1} = u$. By assumption, it is also the density of $\mathbf{z}_{t_2}$ conditionally to $u_{t_2} = u$. Theorem 3 provides sufficient conditions for assumption (A2) to hold:

**Theorem 3** *Assume that $Q$ has full rank, $\min_u \pi(u) > 0$ and the $(\gamma_u)_{1 \leqslant u \leqslant K}$ are linearly independent, then (A2) is satisfied as soon as the functions $(\eta \mapsto \int \exp(\langle \eta, \mathbf{z} \rangle) \gamma_v(\mathbf{z}) d\mathbf{z})_{1 \leqslant v \leqslant K}$ do not have simultaneous zeros.*

Besides the non-simultaneous zeros assumption, the assumptions of Theorem 3 are reminiscent of those used for the identifiability of non-parametric hidden Markov models, see for instance Gassiat et al. (2016); Lehéricy (2019). The key element is that $\mathbf{z}_{t_1}$ and $\mathbf{z}_{t_2}$ are not independent. Thus, we see that (A2) holds if the $\pi$ and the $\gamma$ are not degenerate (in the precise sense given by Theorem 3), for the latent state models in Figs. 1a,1c. Another situation where (A2) holds is when $\mathbf{z}_{t_2}$ is a complete statistic (Lehmann and Casella, 2006) in the statistical model $\{\mathbb{P}_{\mathbf{z}_{t_2}|\mathbf{z}_{t_1}}(\cdot|\mathbf{z}_{t_1})\}_{\mathbf{z}_{t_1}}$, where $\mathbb{P}_{\mathbf{z}_{t_2}|\mathbf{z}_{t_1}}(\cdot|\mathbf{z}_{t_1})$ is the distribution of $\mathbf{z}_{t_2}$ conditionally to $\mathbf{z}_{t_1}$. Consider the two following examples where this holds: 1) When the model $\{\mathbb{P}_{\mathbf{z}_{t_2}|\mathbf{z}_{t_1}}(\cdot|\mathbf{z}_{t_1})\}_{\mathbf{z}_{t_1}}$ is an exponential family. In this situation, complete statistics are known. 2) Autoregressive models with additive innovation of the form $\mathbf{z}_{t_2} = \mathbf{h}(\mathbf{z}_{t_1}) + \mathbf{v}_{t_2}$ for some bijective function $\mathbf{h}$ when the additive noise $\mathbf{v}_{t_2}$ is a complete statistic in the statistical model $\{\mathbb{P}_{\mathbf{v}_{t_2}|\mathbf{z}_{t_1}}(\cdot|\mathbf{z}_{t_1})\}_{\mathbf{z}_{t_1}}$ (note that $\mathbf{v}_{t_2}$ cannot be independent of $\mathbf{z}_{t_1}$ here). The case in Fig. 1b is typically covered by this example.

**Assumption (A3)** states that no direction of the noise free data has a non Dirac Gaussian variable component. It holds as soon as $\mathbf{z}_t = \mathbf{f}(\mathbf{s}_t)$ and the range of $\mathbf{f}$ is such that its orthogonal projection on

any line is not the full line. This assumption holds for instance in the following cases: 1) The range of $\mathbf{f}$ is compact, or 2) the range of $\mathbf{f}$ is contained in a half-cylinder, that is, there exists a hyperplane such that the range of $\mathbf{f}$ is only on one side of this hyperplane and the projection of the range of $\mathbf{f}$ on this hyperplane is bounded.

**Assumption (B1) and Assumption (B2)**  are similar to those in (Hyvärinen and Morioka, 2017; Oberhauser and Schell, 2021) in the special case of time-series, i.e. $\mathbb{T} = \mathbb{N}$. (B1) then entails that there must be sufficiently strong statistical dependence between nearby time points. (B2) is a condition which excludes Gaussian processes and processes which can be trivially transformed to be Gaussian. (For treatment of the Gaussian case, see Appendix B in Supplementary Material.) We can further provide a simple and equivalent formulation when the independent components $\mathbf{s}^{(i)}$ follow independent and stationary HMMs with two hidden states, which is a special case of SNICA. Denote by $\gamma_0^{(i)}$ and $\gamma_1^{(i)}$ the densities of $s_t^{(i)}$ conditionally to $\{u_t^{(i)} = 0\}$ and $\{u_t^{(i)} = 1\}$ respectively.

**Theorem 4** *Assume that the stationary distribution $\pi$ of the hidden chain is such that $0 < \pi(0) < 1$ and that its transition matrix is invertible. Then (B1) and (B2) are satisfied with $m = 2$ if and only if on any open interval, $\gamma_0^{(i)}$ and $\gamma_1^{(i)}$ are not proportional.*

Thus, a very simple HMM leads to these conditions being verified. Hyvärinen and Morioka (2017) already showed that the conditions (B1) and (B2) also hold in the case of non-Gaussian autoregressive models. Thus, we see that our identifiability theory applies both in the case HMM's (Fig 1a) and autoregressive models (Fig 1b), the two principal kinds of temporal structure proposed in previous work, while extending them to further cases and combinations such as in Fig 1c,1d.

**A simplification of (B1,B2)**  It is also possible to combine the assumptions (B1) and (B2) in one, while slightly weakening the generality. The key is to notice that (6) in (B2) implies the derivative in (B1) is zero, by setting $c = 0$. But there is still the difference that (B2) considers all but one index while (B1) considers all indices $i$. If we simply assume (6) does not hold for any $i$, we can replace (B1) and (B2) by the new condition:

- (B') For any open subset $A \subset S^m$ and for any $i \in \{1, \ldots, N\}$, a function $\alpha : \mathbb{R}^{m-1} \to \mathbb{R}$ and a constant $c \in \mathbb{R}$ do not exist such that (6) would hold for all $s \in A$.

Note that Hyvärinen and Morioka (2017) defined uniform dependency and (non-)quasi-Gaussianity as two separate properties, but in fact their assumption of non-quasi-Gaussianity was weaker than ours: it did not consider all open subsets separately, which is why this simplification was not possible for them. We believe their definition of non-quasi-Gaussianity was in fact not quite sufficient to prove their theorem, and our stronger version may be needed, in line with Oberhauser and Schell (2021).

## 5   Experiments

**Estimation method**  One challenge is that it is not practically possible to learn $\Delta$-SNICA by exact maximum-likelihood methods. Instead, we perform learning and inference using Structured VAEs (Johnson et al., 2016) – the current state-of-art in variational inference for structured models. Specifically, this consists of assuming that the latent posterior factorizes as per $q(\mathbf{y}_{1:T}^{(1:N)}, u_{1:T}^{(1:N)}) = \prod_{i=1}^{N} q(\mathbf{y}_{1:T}^{(i)}) q(u_{1:T}^{(i)})$, which allows us to optimize the resulting evidence lower bound (ELBO):

$$\log \widehat{\mathcal{L}} = \mathbb{E}_q \left[ \sum_{t=1}^{T} \log p(\mathbf{x}_t \mid \mathbf{s}_t^{(1)}, ..., \mathbf{s}_t^{(N)}) \right] + \sum_{i=1}^{N} \left( - \text{KL} \left[ q(u_{1:T}^{(i)}) \Big| p(u_{1:T}^{(i)}) \right] + \text{H} \left[ q(\mathbf{s}_{1:T}^{(i)}) \right] \right.$$

$$\left. + \mathbb{E}_q \left[ \log p(\mathbf{s}_1^{(i)} \mid u_1^{(i)}) \right] + \sum_{t=2}^{T} \mathbb{E}_q \left[ \log p(\mathbf{s}_t^{(i)} \mid \mathbf{s}_{t-1}^{(i)}, u_t^{(i)}) \right] \right). \tag{7}$$

Since all the distributions are in conjugate exponential families (encoder neural network is used to approximate the natural parameters of the nonlinear likelihood term) efficient message passing can be used for inference, and the mixing function is learned as decoder neural network. Even though this method lacks consistency guarantees (but see Wang and Blei (2018)), we find that our model performs very well. A more detailed treatment of estimation and inference of $\Delta$-SNICA is given in Appendix C. Our code will be openly available at https://github.com/HHalva/snica.

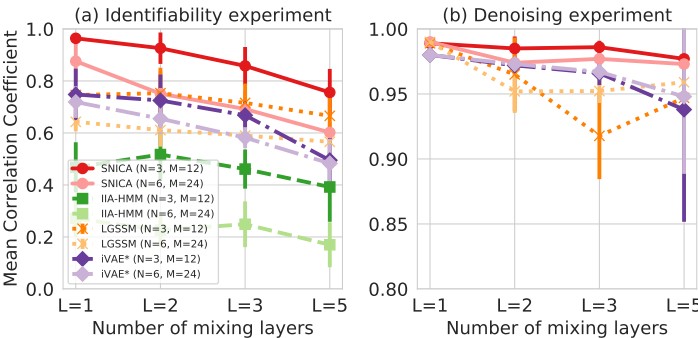

Figure 2: (a) Mean absolute correlation coefficients between ground-truth independent components and their estimates by $\Delta$-SNICA , IIA-HMM, LGSSM and iVAE*, with different orders of complexity (number of layers) and two different dimensions of observed (12, 24) and latent (6, 12) data. (b) Mean absolute correlation coefficient between estimated noise free data and ground-truth noise free data for same set of models except IIA-HMM. Please note the difference in y-axis scales.

## 5.1 Experiments on simulated data

The identifiability theorems stated above hold in the limit of infinite data. Additionally, a consistent estimator would be required to learn the ground-truth components. In the real world, we are limited by data and estimation methods and hence it is unclear as to what extent we are actually able to estimate identifiable components – and whether identifiability reflects in better performance in real world tasks. To explore this, we first performed experiments on simulated data. We compared the performance of our model to the current state-of-the-art, IIA-HMM (Morioka et al., 2021), as well as identifiable VAE (iVAE) (Khemakhem et al., 2020a) and standard linear Gaussian state-space model (LGSSM). Since iVAE is not able to handle latent auxiliary variables, we allow it to "cheat" by giving it access to the true data generating latent-state, thereby creating a presumably challenging baseline (denoted iVAE* in our figures). LGSSM was included as a naive baseline which is only able to estimate linear mixing function.

**Investigating identifiability and consistency**   We simulated 100K long time-sequences from the $\Delta$-SNICA  model and computed the mean absolute correlation coefficient (MCC) between the estimated latent components and ground truth independent components (see Supplementary material for further implementation details).  More precisely, to illustrate the dimensionality reduction capabilities we considered two settings where the observed data dimension $M$, was either 12 or 24 and the number of independent components, $N$ was 3 and 6, respectively.  Since IIA-HMM is unable to do dimensionality reduction, we used PCA to get the data dimension to match that of the latent states.  We considered four levels of mixing of increasing complexity by randomly initialized MLPs of the following number of layers: 1 (linear ICA), 2, 3, and 5. The results in Figure 2a) illustrate the clearly superior performance of our model. The especially poor performance of IIA-HMM maybe explained by lack of noise model, much simpler latent dynamics, and lost information due to PCA pre-processing. See Appendix D for further discussion and training details.

**Application to denoising**   $\Delta$-SNICA is able to denoise time-series signals by learning the generative model and then performing inference on latent variables. Specifically, SVAE learns the encoder network which is used to perform inference on the posterior of the independent components. We illustrate this using the same settings as above, with the exception that we now use our learned encoder and inference to get the posterior means of the independent components and input these in to the estimated decoder to get predicted noise-free observations, denoted as $\widehat{\mathbf{f}}(\mathbf{s}_t)$ – we measured the correlation between $\widehat{\mathbf{f}}(\mathbf{s}_t)$ and the ground-truth $\mathbf{f}(\mathbf{s}_t)$. Note that IIA-HMM, is not able to perform this task. The results in Figure 2b) show that the other models, designed to handle denoising, perform well at this task, as would be expected – identifiability of the latent state is not necessary for good denoising performance. For LGSSM, denoising is done with the Kalman Smoother algorithm.

## 5.2 Experiments on real MEG data

To demonstrate real-data applicability, Δ-SNICA was applied to multivariate time series of electrical activity in the human brain, measured by magnetoencephalography (MEG). Recently, many studies have demonstrated the existence of fast transient networks measured by MEG in the resting state and the dynamic switching between different brain networks (Baker et al., 2014; Vidaurre et al., 2017). Additionally, such MEG data is high-dimensional and very noisy. Thus this data provides an excellent target for Δ-SNICA to disentangle the underlying low-dimensional components.

**Data and Preprocessing**  We considered a resting state MEG sessions from the Cam-CAN dataset. During the resting state recording, subjects sat still with their eyes closed. In the task-session data, the subjects carried out a (passive) audio–visual task including visual stimuli and auditory stimuli. We exclusively used the resting-session data for the training of the network, and task-session data was only used in the evaluation. The modality of the sensory stimulation provided a class label that we used in the evaluation, giving in total two classes. We band-pass filtered the data between 4 Hz and 30 Hz (see Supplementary Material for the details of data and settings).

**Methods**  The resting-state data from all subjects were temporally concatenated and used for training. The number of layers of the decoder and encoder were equal and took values 2, 3, 4. We fixed the number of independent components to 5 so that our result can be fairly compared to those in Morioka et al. (2021). To evaluate the obtained features, we performed classification of the sensory stimulation categories by applying feature extractors trained with (unlabeled) resting-state data to (labeled) task-session data. Classification was performed using a linear support vector machine (SVM) classifier trained on the stimulation modality labels and sliding-window-averaged features obtained for each trial. The performance was evaluated by the generalizability of a classifier across subjects. i.e., one-subject-out cross-validation. For comparison, we evaluated the baseline methods: IIA-HMM and IIA-TCL (Morioka et al., 2021). We also visualized the spatial activity patterns obtained by Δ-SNICA , using the weight vectors from encoder neural network across each layer.

**Results**  Figure 3 a) shows the classification accuracies of the stimulus categories, across different methods and the number of layers for each model. The performances by Δ-SNICA were consistently higher than those by the other (baseline) methods, which indicates the importance of the modeling of the MEG signals by Δ-SNICA . Figure 3 b) shows an example of spatial patterns from the encoder network learned by the Δ-SNICA . We used the visualization method presented in (Hyvärinen and Morioka, 2016). We manually picked one out of the hidden nodes from the third layer in encoder network, and plotted its weighted-averaged sensor signals, We also visualized the most strongly contributing second- and first-layer nodes. We see progressive pooling of L1 units to form left lateral frontal, right lateral frontal and parietal patterns in L2 which are then all pooled together in L3 resulting in a lateral frontoparietal pattern. Most of the spatial patterns in the third layer (not shown) are actually similar to those previously reported using MEG (Brookes et al., 2011). Appendix E provides more detail to the interpretation of the Δ-SNICA results.

## 6 Related work

Previous works on nonlinear ICA have exploited autocorrelations (Hyvärinen and Morioka, 2017; Oberhauser and Schell, 2021) and nonstationarities (Hyvärinen and Morioka, 2016; Hälvä and Hyvärinen, 2020) for identifiability. The SNICA setting provides a unifying framework which allows for both types of temporal dependencies, and further, extends identifiability to other temporal structures as well as any arbitrary higher order data structures which has not previously been considered in the context of nonlinear ICA. Another major theoretical contribution here is to show that identifiability with noise of unknown, arbitrary distribution, while previous work on noisy nonlinear ICA assumed noise of known distribution and known variance (Khemakhem et al., 2020a).

Importantly, the SNICA framework is fully probabilistic and thus accommodates higher order latent variables, leading to "purely unsupervised" learning. This is in large contrast to previous research which have been developed for the case where we are able to observe some additional auxiliary variable, such as audio signals accompanying video (Hyvärinen et al., 2019; Khemakhem et al., 2020a,b), or heuristically define the auxiliary variable based on time structure (Hyvärinen and Morioka, 2016). In practice this means that we are able to estimate our models using (variational)

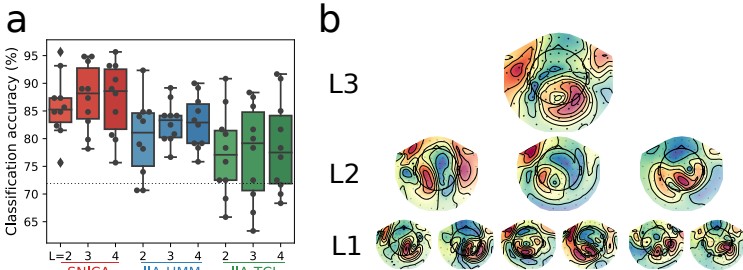

Figure 3: Δ-SNICA on MEG data. (a) Classification accuracies of linear SVMs trained with auditory-visual data to predict stimulus category, with feature extractors trained by Δ-SNICA in advance with resting-state data. Each point represents a testing accuracy on a target subject (chance level: 50%). Horizontal dotted line is PCA-only baseline. (b) Example of spatial patterns of the components learned by Δ-SNICA (L=3). Each topography corresponds to one spatial pattern. L3: approximate total spatial pattern of one third-layer unit. L2: the patterns of the three second-layer units maximally contributing to this L3 unit. L1: for each L2 unit, the two most strongly contributing first-layer units.

MLE, which is more principled than the heuristic self-supervised methods in most earlier papers. The only existing frameworks allowing MLE (Hälvä and Hyvärinen, 2020; Khemakhem et al., 2020a) used model restricted to exponential families, and had either no HMM or a very simple one.

The switching linear dynamical model, Δ-SNICA in Section 3.2, shows the above benefits in the form of a single model. That is, unlike previous nonlinear ICA models, it combines: 1) temporal dependencies and "non-stationarity" (or HMM) in a single model 2) dimensionality reduction within a rigorous maximum likelihood learning and inference framework, and 3) a separate observation equation with general observational noise. This results in a very rich, realistic, and principled model for time series.

Very recently, Morioka et al. (2021) proposed a related model by considering innovations of time series to be nonstationary. However, their model is noise-free, restricted to exponential families of at least order two, and not applicable to the spatial case, thus making our identifiability results significantly stronger. From a more practical viewpoint, their model suffers from the fact that it either does not allow for dimensionality reduction (if an HMM is used) or requires a manual segmentation (if HMM is not used). Nor does it have a clear distinction into a state dynamics equation and a measurement equation which allows for cleaning or denoising of the data.

**Limitations** Our identifiability theory makes some restrictive assumptions, and it remains to be seen if they could be lifted in future work. In particular, the data is not allowed to have too heavy tails; the noise must be additive, and independent of the signal; and the practical interpretation of some of the assumptions, such as (A3) is difficult. It is also difficult to say whether our assumption of unconditionally independent components is realistic in practice. Regarding practical applications, our specific model only scratches the surface of what is possible in this framework. In particular, we did not develop a model with spatial distributions, nor did we model non-Gaussian observational noise – our main aim was to lay the foundations for the relevant identification theory. Future work should aim to make the estimation more efficient computationally; this is a ubiquitous problem in deep learning, but specific solutions for this concrete problem may be achievable (Gresele et al., 2020).

## 7 Conclusion

We proposed a new general framework for identifiable disentanglement, based on nonlinear ICA with very general temporal dynamics or spatial structure. Observational noise of arbitrary unknown distribution is further included. We prove identifiability of the models in this framework with high generality and mathematical rigour. For real data analysis, we propose a special case which subsumes the properties of all existing time series models in nonlinear ICA, while generalizing them in many ways (see Section 6 for details). We hope this work will contribute to wide-spread application of identifiable methods for disentanglement in a highly principled, probabilistic framework.

