## Acknowledgments and Disclosure of Funding

The authors would like to thank Richard Turner for insightful comments and discussion on this work. The authors also wish to thank the Finnish Grid and Cloud Infrastructure (FGCI) for supporting this project with computational and data storage resources. A.H. was supported by a Fellowship from CIFAR, and the Academy of Finland. E.G. would like to acknowledge support for this project from Institut Universitaire de France. J.S. is supported by the University of Cambridge Harding Distinguished Postgraduate Scholars Programme.

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

# A Appendix

## A.1 Proof of Theorem 1

Let $m \geqslant 2$ and $(t_1, \ldots, t_m) \in \mathbb{T}^m$. Let $R_m$ and $\tilde{R}_m$ be two possible distributions for $(\mathbf{z}_{t_1}, \ldots, \mathbf{z}_{t_m})$ that satisfy assumptions (A1), (A2) and (A3) and let $P$ and $\tilde{P}$ be two possible distributions for $\varepsilon_{t_1}$. Assume that the distribution of $(\mathbf{x}_{t_1}, \ldots, \mathbf{x}_{t_m})$ in the model (5) is the same under $(R_m, P)$ and $(\tilde{R}_m, \tilde{P})$.

Write $\Phi_{R_m}$ the characteristic function of $R_m$, and likewise $\Phi_{\tilde{R}_m}$, $\Phi_P$ and $\Phi_{\tilde{P}}$. Following the proof of Theorem 1 of Gassiat et al. (2020b) on the distribution of $(\mathbf{z}_{t_1}, \mathbf{z}_{t_2})$, as in Assumption (A1) we have $\rho < 3$, by Hadamard's factorization theorem, there exist a polynomial function $Q$ with total degree at most 2 and a neighborhood $V$ of 0 in $\mathbb{R}^M$ such that for all $\mathbf{u} \in V$,

$$\Phi_P(\mathbf{u}) \exp\{Q(\mathbf{u})\} = \Phi_{\tilde{P}}(\mathbf{u}). \tag{8}$$

For completeness we provide at the end of this section the sketch of proof of (8).

Writing the characteristic function of $(\mathbf{z}_{t_1}, \ldots, \mathbf{z}_{t_m})$ under the two sets of parameters yields, for all $(\mathbf{u}_1, \ldots, \mathbf{u}_m) \in V^m$,

$$\Phi_{R_m}(\mathbf{u}_1, \ldots, \mathbf{u}_m) \prod_{k=1}^m \Phi_P(\mathbf{u}_k) = \Phi_{\tilde{R}_m}(\mathbf{u}_1, \ldots, \mathbf{u}_m) \left( \prod_{k=1}^m \Phi_P(\mathbf{u}_k) \right) \left( \prod_{k=1}^m \exp(Q(\mathbf{u}_k)) \right). \tag{9}$$

Since $\Phi_P$ is continuous and non-zero at 0, we may divide both sides by $\prod_{k=1}^m \Phi_P(\mathbf{u}_k)$ on a neighborhood of zero. Under assumption (A1), $\Phi_{R_m}$ and $\Phi_{\tilde{R}_m}$ can be extended into multivariate analytic functions:

$$\Phi_{R_m}: \qquad (\mathbb{C}^M)^m \quad \longrightarrow \quad \mathbb{C}$$

$$(\mathbf{u}_1, \ldots, \mathbf{u}_m) \quad \longmapsto \quad \int \exp\left( i\mathbf{u}_1^\top \mathbf{z}_{t_1} + \cdots + i\mathbf{u}_m^\top \mathbf{z}_{t_m} \right) \mathrm{d}R_m(\mathbf{z}_{t_1}, \ldots, \mathbf{z}_{t_m}).$$

We will need the following statement used in Gassiat et al. (2020a) and Gassiat et al. (2020b). We provide a proof at the end of the section for completeness, see also Shabat (1992).

**Lemma 1** *If a multivariate function is analytic on the whole multivariate complex space and is the null function on an open set of the multivariate real space or on an open set of the multivariate purely imaginary space, then it is the null function on the whole multivariate complex space.*

Thus, equation (9) can be extended on $(\mathbb{C}^M)^m$, which shows that for all $(\mathbf{u}_1, \ldots, \mathbf{u}_m) \in (\mathbb{C}^M)^m$,

$$\Phi_{R_m}(\mathbf{u}_1, \ldots, \mathbf{u}_m) = \Phi_{\tilde{R}_m}(\mathbf{u}_1, \ldots, \mathbf{u}_m) \prod_{k=1}^m \exp\{Q(\mathbf{u}_k)\}.$$

As $\Phi_{R_m}$ and $\Phi_{\tilde{R}_m}$ are characteristic functions, $Q$ has no constant term. The degree 1 term corresponds to a translation parameter. Without loss of generality, assume that $\mathbf{z}_{t_1}$ is centered under $R_m$ and $\tilde{R}_m$, then

$$i \, \mathbb{E}_{R_m}[\mathbf{z}_{t_1}] = \nabla_{\mathbf{u}_1} \Phi_{R_m}(0) = \nabla_{\mathbf{u}_1} \Phi_{\tilde{R}_m}(0) + \nabla Q(0) = i \, \mathbb{E}_{\tilde{R}_m}[\mathbf{z}_{t_1}] + \nabla Q(0),$$

which entails $\nabla Q(0) = 0$. Thus, $Q$ only has terms of degree 2, which means it is a quadratic form in $\mathbb{R}^M$. Writing $Q(\mathbf{u}) = \mathbf{u}^\top (Q_+ - Q_-)\mathbf{u}$ where $Q_+$ and $Q_-$ are the positive semi-definite matrices corresponding to the positive and negative eigenvalues of $Q$ respectively, yields

$$\Phi_{R_m}(\mathbf{u}_1, \ldots, \mathbf{u}_m) \prod_{k=1}^m \exp\left\{ -\mathbf{u}_k^\top Q_+ \mathbf{u}_k \right\} = \Phi_{\tilde{R}_m}(\mathbf{u}_1, \ldots, \mathbf{u}_m) \prod_{k=1}^m \exp\left\{ -\mathbf{u}_k^\top Q_- \mathbf{u}_k \right\}.$$

From this decomposition, we deduce that if $\mathbf{z} \sim R_m$, $\tilde{\mathbf{z}} \sim \tilde{R}_m$, and $(\mathbf{v}_k)_{1 \leqslant k \leqslant m}$ (resp. $(\tilde{\mathbf{v}}_k)_{1 \leqslant k \leqslant m}$) are i.i.d. multivariate Gaussian random variables with mean 0 and covariance matrices $2Q_+$ (resp. $2Q_-$) that are independent of $\mathbf{z}$ (resp. $\tilde{\mathbf{z}}$), then $(\mathbf{z}_{t_k} + \mathbf{v}_k)_{1 \leqslant k \leqslant m}$ has the same distribution as $(\tilde{\mathbf{z}}_{t_k} + \tilde{\mathbf{v}}_k)_{1 \leqslant k \leqslant m}$. In particular, the supports of the $\mathbf{v}_k$, $1 \leqslant k \leqslant m$ and of the $\tilde{\mathbf{v}}_k$, $1 \leqslant k \leqslant m$, are orthogonal.

Let $\Pi_-$ be the orthogonal projection on the support of $\tilde{\mathbf{v}}_k$, then $\Pi_- \mathbf{z}_{t_k} = \Pi_- \tilde{\mathbf{z}}_{t_k} + \tilde{\mathbf{v}}_k$, which by assumption (A3) entails $Q_- = 0$ (otherwise, take a non-zero $\eta$ in the support of $\tilde{\mathbf{v}}_k$). Since $\tilde{\mathbf{z}}$ satisfies the same assumptions as $\mathbf{z}$, $Q_+ = 0$ for the same reason. Thus, $Q = 0$, so that $\Phi_{R_m} = \Phi_{\tilde{R}_m}$, and then $R_m = \tilde{R}_m$, and likewise $P = \tilde{P}$.

**Proof of** (8). Since the distribution of $(\mathbf{x}_{t_1}, \mathbf{x}_{t_2})$ in the model (5) is the same under $(R_2, P)$ and $(\tilde{R}_2, \tilde{P})$ (likewise for the distribution of $\mathbf{x}_t$ under $(R_1, P)$ and $(\tilde{R}_1, \tilde{P})$ for any $t$), we get that for all $\mathbf{u} \in \mathbb{R}^M$,

$$\Phi_P(\mathbf{u})\Phi_{R_1}(\mathbf{u}) = \Phi_{\tilde{P}}(\mathbf{u})\Phi_{\tilde{R}_1}(\mathbf{u}) \tag{10}$$

and for all $(\mathbf{u}_1, \mathbf{u}_2) \in (\mathbb{R}^M)^2$,

$$\Phi_P(\mathbf{u}_1)\Phi_P(\mathbf{u}_2)\Phi_{R_2}(\mathbf{u}_1, \mathbf{u}_2) = \Phi_{\tilde{P}}(\mathbf{u}_1)\Phi_{\tilde{P}}(\mathbf{u}_2)\Phi_{\tilde{R}_2}(\mathbf{u}_1, \mathbf{u}_2). \tag{11}$$

There exists a neighborhood $W$ of 0 in $\mathbb{R}^M$ such that $\Phi_P$ and $\Phi_{\tilde{P}}$ do not vanish on $W$, so that equations (10) and (11) give that for all $(\mathbf{u}_1, \mathbf{u}_2) \in W^2$,

$$\Phi_{R_2}(\mathbf{u}_1, \mathbf{u}_2)\Phi_{\tilde{R}_1}(\mathbf{u}_1)\Phi_{\tilde{R}_1}(\mathbf{u}_2) = \Phi_{\tilde{R}_2}(\mathbf{u}_1, \mathbf{u}_2)\Phi_{R_1}(\mathbf{u}_1)\Phi_{R_1}(\mathbf{u}_2). \tag{12}$$

Application of Lemma 1 yields now that (12) holds for all $(\mathbf{u}_1, \mathbf{u}_2) \in (\mathbb{C}^M)^2$. Using Assumption (A2) and Lemma 1 we easily deduce from (12) that the set of zeros of $\Phi_{R_1}$ and $\Phi_{\tilde{R}_1}$ are equal. Then, using Assumption (A1) and Hadamard's factorization Theorem, see Stein and Shakarchi (2003) (Chapter 5 Theorem 5.1), and arguing variable by variable, we deduce that there exists a function $Q$ on $\mathbb{C}^M$ such that, for all $i = 1, \ldots, M$, $Q$ is a polynomial function with degree at most 2 (and coefficients depending on $(u^{(1)}, \ldots, u^{(i-1)}, u^{(i+1)}, \ldots, u^{(M)})$) and for all $\mathbf{u} = (u^{(1)}, \ldots, u^{(M)}) \in \mathbb{C}^M$,

$$\Phi_{R_1}(\mathbf{u}) = \Phi_{\tilde{R}_1}(\mathbf{u})\exp(Q(\mathbf{u})).$$

Using again Assumption (A1) allows to deduce that $Q$ has total degree 2. Coming back to equation (10) yields for all $\mathbf{u} \in \mathbb{R}^M$,

$$\Phi_P(\mathbf{u})\Phi_{\tilde{R}_1}(\mathbf{u})\exp(Q(\mathbf{u})) = \Phi_{\tilde{P}}(\mathbf{u})\Phi_{\tilde{R}_1}(\mathbf{u}) \tag{13}$$

which, on the neighborhood $V$ of 0 in $\mathbb{R}^M$ where $\Phi_{\tilde{R}_1}$ does not vanish, proves (8).

**Proof of Lemma 1** We prove the statement by induction on the number $d$ of variables. If $h$ is analytic on $\mathbb{C}$ and is not the null function, then $h$ has isolated zeros, so that Lemma 1 holds for $d = 1$. Assume that the lemma holds for analytic functions on $\mathbb{C}^d$ and let $h$ be an analytic function on $\mathbb{C}^{d+1}$ which is the null function on an open set $A$ of $\mathbb{R}^{d+1}$. Then, there exists open sets $B_1, \ldots, B_{d+1}$ of $\mathbb{R}$ such that $B_1 \times \cdots \times B_{d+1} \subset A$. For any $t \in B_{d+1}$, let $h_t : \mathbb{C}^d \to \mathbb{C}$ such that $h_t(\cdot) = h(\cdot, t)$, then $h_t$ is analytic on $\mathbb{C}^d$ and is the null function on $B_1 \times \cdots \times B_d$ so that by the induction hypothesis, for all $z \in \mathbb{C}^d$, $h_t(z) = 0$, that is $h(z, t) = 0$ for all $z \in \mathbb{C}$ and for all $t \in B_{d+1}$. Therefore, for any $z \in \mathbb{C}^d$, the function $h(z, \cdot)$ is analytic on $\mathbb{C}$ and is the null function on $B_{d+1}$ so that for any $z_0 \in \mathbb{C}$, $h(z, z_0) = 0$ and $h$ is the null function. The proof when $h$ is the null function on an open set of the multivariate purely imaginary space is similar.

## A.2  Proof of Theorem 2

In the following, the index $m$ may be dropped in the notations $p_m^{(i)}$ and $Q_m^{(i)}$ when there is no confusion. Let $p^{(i)}$, $\tilde{p}^{(i)}$, $\mathbf{f}$ and $\tilde{\mathbf{f}}$ be such that if $\mathbf{s} \sim p^{(i)}$ and $\tilde{\mathbf{s}} \sim \tilde{p}^{(i)}$, then $\mathbf{f}(\mathbf{s})$ and $\tilde{\mathbf{f}}(\tilde{\mathbf{s}})$ have the same distribution. Write $\mathbf{g} = \mathbf{f}^{-1}$ and $\tilde{\mathbf{g}} = \tilde{\mathbf{f}}^{-1}$.

Let $\mathbf{x}_1, \ldots, \mathbf{x}_m \in \mathcal{M}$. For each $k \in \{1, \ldots, m\}$, let $\vartheta_k \in \Theta$ such that $\mathbf{x}_k \in U_{\vartheta_k}$ and let $\mathbf{w}_k = \varphi_{\vartheta_k}(\mathbf{x}_k)$. Writing the density of the random vector $(\varphi_{\vartheta_1}(\mathbf{f}(\mathbf{s}_{t_1})), \ldots, \varphi_{\vartheta_m}(\mathbf{f}(\mathbf{s}_{t_m})))$ at $(\mathbf{w}_1, \ldots, \mathbf{w}_m)$ with respect to the Lebesgue measure for the two parameterizations, yields

$$\prod_{k=1}^{m} |J_{\mathbf{g}\circ\varphi_{\vartheta_j}^{-1}}(\mathbf{w}_k)| \prod_{i=1}^{N} p^{(i)}((\mathbf{g}^{(i)} \circ \varphi_{\vartheta_1}^{-1})(\mathbf{w}_1), \ldots, (\mathbf{g}^{(i)} \circ \varphi_{\vartheta_m}^{-1})(\mathbf{w}_m))$$

$$= \prod_{k=1}^{m} |J_{\tilde{\mathbf{g}}\circ\varphi_{\vartheta_k}^{-1}}(\mathbf{w}_k)| \prod_{i=1}^{N} \tilde{p}^{(i)}((\tilde{\mathbf{g}}^{(i)} \circ \varphi_{\vartheta_1}^{-1})(\mathbf{w}_1), \ldots, (\tilde{\mathbf{g}}^{(i)} \circ \varphi_{\vartheta_m}^{-1})(\mathbf{w}_m)). \tag{14}$$

Let $k, \ell \in \{1, \ldots, m\}$ and $u, v \in \{1, \ldots, N\}$ be such that $k \neq \ell$, then by (14),

$$\sum_{i=1}^{N} \frac{\partial^2}{\partial w_k^{(u)} \partial w_\ell^{(v)}} \log p^{(i)}((\mathbf{g}^{(i)} \circ \varphi_{\vartheta_1}^{-1})(\mathbf{w}_1), \ldots, (\mathbf{g}^{(i)} \circ \varphi_{\vartheta_m}^{-1})(\mathbf{w}_m))$$

$$= \sum_{i=1}^{N} \frac{\partial^2}{\partial w_k^{(u)} \partial w_\ell^{(v)}} \log \tilde{p}^{(i)}((\tilde{\mathbf{g}}^{(i)} \circ \varphi_{\vartheta_1}^{-1})(\mathbf{w}_1), \ldots, (\tilde{\mathbf{g}}^{(i)} \circ \varphi_{\vartheta_m}^{-1})(\mathbf{w}_m)),$$

that is

$$\sum_{i=1}^{N} \frac{\partial^2 \log p^{(i)}}{\partial s_k^{(i)} \partial s_\ell^{(i)}} \left( (\mathbf{g}^{(i)} \circ \varphi_{\vartheta_1}^{-1})(\mathbf{w}_1), \ldots, (\mathbf{g}^{(i)} \circ \varphi_{\vartheta_m}^{-1})(\mathbf{w}_m) \right) \frac{\partial(\mathbf{g}^{(i)} \circ \varphi_{\vartheta_k}^{-1})}{\partial w^{(u)}} (\mathbf{w}_k) \frac{\partial(\mathbf{g}^{(i)} \circ \varphi_{\vartheta_\ell}^{-1})}{\partial w^{(v)}} (\mathbf{w}_\ell)$$

$$= \sum_{i=1}^{N} \frac{\partial^2 \log \tilde{p}^{(i)}}{\partial s_k^{(i)} \partial s_\ell^{(i)}} \left( (\tilde{\mathbf{g}}^{(i)} \circ \varphi_{\vartheta_1}^{-1})(\mathbf{w}_1), \ldots, (\tilde{\mathbf{g}}^{(i)} \circ \varphi_{\vartheta_m}^{-1})(\mathbf{w}_m) \right) \frac{\partial(\tilde{\mathbf{g}}^{(i)} \circ \varphi_{\vartheta_k}^{-1})}{\partial w^{(u)}} (\mathbf{w}_k) \frac{\partial(\tilde{\mathbf{g}}^{(i)} \circ \varphi_{\vartheta_\ell}^{-1})}{\partial w^{(v)}} (\mathbf{w}_\ell).$$

For all $(\mathbf{s}_1, \ldots, \mathbf{s}_m) \in S^m$, let

$$q_{i,(k,\ell)} = \frac{\partial^2 \log p^{(i)}}{\partial s_k^{(i)} \partial s_\ell^{(i)}}, \quad \tilde{q}_{i,(k,\ell)} = \frac{\partial^2 \log \tilde{p}^{(i)}}{\partial s_k^{(i)} \partial s_\ell^{(i)}},$$

$$D_{k,\ell}(\mathbf{s}_1, \ldots, \mathbf{s}_m) = \operatorname{diag}\left( q_{i,(k,\ell)}\left( s_1^{(i)}, \ldots, s_m^{(i)} \right) \right)_{1 \leqslant i \leqslant N},$$

$$\tilde{D}_{k,\ell}(\mathbf{s}_1, \ldots, \mathbf{s}_m) = \operatorname{diag}\left( \tilde{q}_{i,(k,\ell)}\left( (\tilde{\mathbf{g}}^{(i)} \circ \mathbf{g}^{-1})(\mathbf{s}_1), \ldots, (\tilde{\mathbf{g}}^{(i)} \circ \mathbf{g}^{-1})(\mathbf{s}_m) \right) \right)_{1 \leqslant i \leqslant N},$$

so that, writing $(J_a)_{ij} = \partial a_i / \partial x_j$ the Jacobian matrix of the map $a$ and $\mathbf{s}_j = \mathbf{g}(\mathbf{x}_j)$ for each $j \in \{1, \ldots, m\}$,

$$J_{\mathbf{g} \circ \varphi_{\vartheta_k}^{-1}}(\mathbf{w}_k)^\top D_{k,\ell}(\mathbf{s}_1, \ldots, \mathbf{s}_m) J_{\mathbf{g} \circ \varphi_{\vartheta_\ell}^{-1}}(\mathbf{w}_\ell) = J_{\tilde{\mathbf{g}} \circ \varphi_{\vartheta_k}^{-1}}(\mathbf{w}_k)^\top \tilde{D}_{k,\ell}(\mathbf{s}_1, \ldots, \mathbf{s}_m) J_{\tilde{\mathbf{g}} \circ \varphi_{\vartheta_\ell}^{-1}}(\mathbf{w}_\ell).$$

Note that for all $\mathbf{w} \in \varphi_{\vartheta_k}(U_{\vartheta_k})$,

$$J_{\tilde{\mathbf{g}} \circ \varphi_{\vartheta_k}^{-1}}(\mathbf{w}) (J_{\mathbf{g} \circ \varphi_{\vartheta_k}^{-1}}(\mathbf{w}))^{-1} = J_{\tilde{\mathbf{g}} \circ \mathbf{g}^{-1}}((\mathbf{g} \circ \varphi_{\vartheta_k}^{-1})(\mathbf{w})),$$

so that for all $(\mathbf{s}_1, \ldots, \mathbf{s}_m) \in S^m$,

$$D_{k,\ell}(\mathbf{s}_1, \ldots, \mathbf{s}_m) = J_{\tilde{\mathbf{g}} \circ \mathbf{g}^{-1}}(\mathbf{s}_k)^\top \tilde{D}_{k,\ell}(\mathbf{s}_1, \ldots, \mathbf{s}_m) J_{\tilde{\mathbf{g}} \circ \mathbf{g}^{-1}}(\mathbf{s}_\ell). \tag{15}$$

Consider the following assertion.

- (P) For all $\mathbf{s}$ in a dense subset of $S$, there exist integers $k, \ell \in \{1, \ldots, m\}$ with $k \neq \ell$ and $\mathbf{s}_1, \ldots, \mathbf{s}_{k-1}, \mathbf{s}_{k+1}, \ldots, \mathbf{s}_m \in S$ such that all entries of the vector

$$\left( \frac{q_{i,(k,\ell)}(\ldots, s^{(i)}, \ldots, s^{(i)}, \ldots) q_{i,(k,\ell)}(\ldots, s_\ell^{(i)}, \ldots, s_\ell^{(i)}, \ldots)}{q_{i,(k,\ell)}(\ldots, s^{(i)}, \ldots, s_\ell^{(i)}, \ldots)^2} \right)_{1 \leqslant i \leqslant N}$$

  are distinct ($s^{(i)}$ and $s_\ell^{(i)}$ are in the positions $k$ and $\ell$ in the equation above).

Assume that (P) holds. [We shall prove below that (P) holds under the assumptions of Theorem 2]. Let $\mathbf{s} = (\mathbf{s}_1, \ldots, \mathbf{s}_m) \in S$ such that $D_{k,\ell}(\mathbf{s}_1, \ldots, \mathbf{s}_m)$ is invertible (any $\mathbf{s}$ in a dense subset of $S$ works thanks to assumption B1). For ease of notation in the following sequence of equations, we drop all unused subscripts and parameters, thus writing $J(\mathbf{s}_k)$ instead of $J_{\tilde{\mathbf{g}} \circ \mathbf{g}^{-1}}(\mathbf{s}_k)$ and $D(\mathbf{s}_k, \mathbf{s}_\ell)$ instead of $D_{k,\ell}(\mathbf{s}_1, \ldots, \mathbf{s}_k, \ldots, \mathbf{s}_\ell, \ldots, \mathbf{s}_m)$ (and likewise for $\tilde{J}$ and $\tilde{D}$). We follow the arguments of the proof of Lemma 2 in Hyvärinen and Morioka (2017) to deduce from (15) an eigenvalue decomposition. Write (15) for several parameters:

$$D(\mathbf{s}_k, \mathbf{s}_k) = J(\mathbf{s}_k)^\top \tilde{D}(\mathbf{s}_k, \mathbf{s}_k) J(\mathbf{s}_k),$$

$$D(\mathbf{s}_k, \mathbf{s}_\ell) = J(\mathbf{s}_k)^\top \tilde{D}(\mathbf{s}_k, \mathbf{s}_\ell) J(\mathbf{s}_\ell)$$

$$= J(\mathbf{s}_\ell)^\top \tilde{D}(\mathbf{s}_k, \mathbf{s}_\ell) J(\mathbf{s}_k) \quad \text{by symmetry,}$$

$$D(\mathbf{s}_\ell, \mathbf{s}_\ell) = J(\mathbf{s}_\ell)^\top \tilde{D}(\mathbf{s}_\ell, \mathbf{s}_\ell) J(\mathbf{s}_\ell),$$

which altogether entails

$$D(\mathbf{s}_k, \mathbf{s}_\ell)^{-1} D(\mathbf{s}_\ell, \mathbf{s}_\ell) D(\mathbf{s}_k, \mathbf{s}_\ell)^{-1} D(\mathbf{s}_k, \mathbf{s}_k)$$

$$= J(\mathbf{s}_k)^{-1} \left[ \tilde{D}(\mathbf{s}_k, \mathbf{s}_\ell)^{-1} \tilde{D}(\mathbf{s}_\ell, \mathbf{s}_\ell) \tilde{D}(\mathbf{s}_k, \mathbf{s}_\ell)^{-1} \tilde{D}(\mathbf{s}_k, \mathbf{s}_k) \right] J(\mathbf{s}_k).$$

The vector in assertion (P) contains the diagonal entries of this diagonal matrix. If they are all distinct, the eigenvalue decomposition is unique, which means that $J(\mathbf{s}_k)$ is the product of a permutation matrix and a diagonal matrix.

Thus, $J_{\tilde{\mathbf{g}} \circ \mathbf{g}^{-1}}$ is the product of a permutation matrix with a diagonal matrix on a dense subset of $S$, and hence on $S$ by regularity of $\mathbf{g}$ and $\tilde{\mathbf{g}}$.

For any permutation matrix $P$, the set of all $\mathbf{s} \in S$ where $J_{\tilde{\mathbf{g}} \circ \mathbf{g}^{-1}}(\mathbf{s})$ is the product of $P$ with an invertible diagonal matrix $D(\mathbf{s})$ is both open (by continuity of $J_{\tilde{\mathbf{g}} \circ \mathbf{g}^{-1}}$) and closed (if $\mathbf{s}_n \to \mathbf{s}$ are such that $J_{\tilde{\mathbf{g}} \circ \mathbf{g}^{-1}}(\mathbf{s}_n) = PD_n$ for all $n$, then by continuity the permutation matrix at $\mathbf{s}$ is also $P$ and since the jacobian is always invertible by the diffeomorphism assumption, $\lim_n D_n$ exists and is invertible). Thus, by connexity of $S$, the permutation is the same for all $\mathbf{s} \in S$. For the next paragraph, we assume without loss of generality that it is the identity permutation.

Therefore, since for all $j$ and $s^{(j)} \in S^{(j)}$, the set $S^{(1)} \times \cdots \times S^{(j-1)} \times \{s_j\} \times S^{(j+1)} \times \cdots \times S^{(N)}$ is connected, $(\tilde{\mathbf{g}} \circ \mathbf{g}^{-1})^{(j)}$ is constant on this set, and thus it depends on $s^{(j)}$ only. It is bijective on $S^{(j)}$ because both $\mathbf{g}$ and $\tilde{\mathbf{g}}$ are. Thus, $\mathbf{g} = \tilde{\mathbf{g}}$ up to a permutation of the coordinates and a bijective transformation of each coordinate.

Let us now prove that assertion (P) is true. The negation of (P) is that there exists an open set $A \subset S$ such that for all $\mathbf{s} \in A$, for all $k, \ell \in \{1, \ldots, m\}$ with $k \neq \ell$ and for all $(\mathbf{s}_1, \ldots, \mathbf{s}_{k-1}, \mathbf{s}_{k+1}, \ldots, \mathbf{s}_m) \in S^{m-1}$, there exists $i, j \in \{1, \ldots, N\}$ with $i \neq j$ such that

$$\frac{q_{i,(k,\ell)}(\ldots, s^{(i)}, \ldots, s^{(i)}, \ldots) q_{i,(k,\ell)}(\ldots, s_\ell^{(i)}, \ldots, s_\ell^{(i)}, \ldots)}{q_{i,(k,\ell)}(\ldots, s^{(i)}, \ldots, s_\ell^{(i)}, \ldots)^2}$$
$$= \frac{q_{j,(k,\ell)}(\ldots, s^{(j)}, \ldots, s^{(j)}, \ldots) q_{j,(k,\ell)}(\ldots, s_\ell^{(j)}, \ldots, s_\ell^{(j)}, \ldots)}{q_{j,(k,\ell)}(\ldots, s^{(j)}, \ldots, s_\ell^{(j)}, \ldots)^2} . \quad (16)$$

Let $\mathbf{s} \in A$, $k, \ell \in \{1, \ldots, m\}$ with $k \neq \ell$. For all $(i, j) \in \{1, \ldots, N\}^2$ with $i \neq j$, define $\tilde{S}_{i,j}$ the subset of $S^{m-1}$ such that for all $(\mathbf{s}_1, \ldots, \mathbf{s}_{k-1}, \mathbf{s}_{k+1}, \ldots, \mathbf{s}_m) \in \tilde{S}_{i,j}$, equation (16) holds. Since the sets $\tilde{S}_{i,j}$, $(i, j) \in \{1, \ldots, N\}^2$, $i \neq j$, form a partition of $S^{m-1}$, which has non-empty interior, there exists at least one pair $(i, j)$ such that the closure of $\tilde{S}_{i,j}$ contains a non-empty open subset $O_{i,j}$. Since $q_{i,(k,\ell)}$ and $q_{j,(k,\ell)}$ are non zero almost everywhere by the uniform $(k, \ell)$-dependency assumption, we may assume without loss of generality that the denominators of equation (16) are non zero for all $(\mathbf{s}_1, \ldots, \mathbf{s}_{k-1}, \mathbf{s}_{k+1}, \ldots, \mathbf{s}_m) \in O_{i,j}$. Thus, by continuity of $q_{i,(k,\ell)}$ and $q_{j,(k,\ell)}$, the terms of equation (16) do not depend on the choice of element in $O_{i,j}$: write $f_{i,(k,\ell)}(s^{(i)}, O_{i,j})$ the left hand term and $f_{j,(k,\ell)}(s^{(j)}, O_{i,j})$ the right hand term.

Let $k, \ell \in \{1, \ldots, m\}$ with $k \neq \ell$. Let $(V_n)_{n \geqslant 1}$ be a basis of open sets of $(\mathbb{R}^N)^{m-1}$. For all $(i, j) \in \{1, \ldots, N\}$ with $i \neq j$ and $n \in \mathbb{N}^*$, let $A_{(i,j),n}$ be the subset of $A$ such that for all $\mathbf{s} \in A_{(i,j),n}$ and all $(\mathbf{s}_1, \ldots, \mathbf{s}_{k-1}, \mathbf{s}_{k+1}, \ldots, \mathbf{s}_m) \in V_n$, equation (16) holds. Then, $A = \bigcup_{n \geqslant 1} \bigcup_{i \neq j} A_{(i,j),n}$ (since $O_{i,j}$ contains at least one of the sets of the basis $(V_n)_{n \geqslant 1}$) and thus there exists $i \neq j$ and $n$ such that the interior of the closure of $A_{(i,j),n}$ is non-empty (otherwise $A$ would be a meagre set and thus have empty interior by Baire's category theorem, which is absurd since $A$ is a non-empty open set). Let $i, j, n$ be such that the closure of $A_{(i,j),n}$ has non-empty interior, and $B$ be a non-empty subset of the closure of $A_{(i,j),n}$. Since $q_{i,(k,\ell)}$ and $q_{j,(k,\ell)}$ are non zero almost everywhere by the uniform $(k, \ell)$-dependency assumption, we may take an open set $V \subset V_n$ and assume without loss of generality that the denominators of equation (16) are non zero for all $(\mathbf{s}_1, \ldots, \mathbf{s}_{k-1}, \mathbf{s}_{k+1}, \ldots, \mathbf{s}_m) \in V$ and all $\mathbf{s} \in B$. Thus, by continuity of $q_{i,(k,\ell)}$ and $q_{j,(k,\ell)}$, the terms of equation (16) do not depend on the choice of element in $B$ or $V$.

To summarize, this means that for all $k, \ell \in \{1, \ldots, m\}$ with $k \neq \ell$, there exists $(i, j) \in \{1, \ldots, N\}$ with $i \neq j$, a constant $c$ and an open set $A' \subset S^m$ such that for all $\mathbf{s} = (\mathbf{s}_1, \ldots, \mathbf{s}_m) \in A'$,

$$q_{i,(k,\ell)}(\ldots, s_k^{(i)}, \ldots, s_\ell^{(i)}, \ldots)^2 = c q_{i,(k,\ell)}(\ldots, s_k^{(i)}, \ldots, s_k^{(i)}, \ldots) q_{i,(k,\ell)}(\ldots, s_\ell^{(i)}, \ldots, s_\ell^{(i)}, \ldots),$$
$$q_{j,(k,\ell)}(\ldots, s_k^{(j)}, \ldots, s_\ell^{(j)}, \ldots)^2 = c q_{j,(k,\ell)}(\ldots, s_k^{(j)}, \ldots, s_k^{(j)}, \ldots) q_{j,(k,\ell)}(\ldots, s_\ell^{(j)}, \ldots, s_\ell^{(j)}, \ldots).$$

This situation is excluded by the local $(k, \ell)$-non quasi Gaussianity assumption, therefore the negation of (P) is false, therefore $\mathbf{g} = \tilde{\mathbf{g}}$ up to permutation and bijective transformation of each coordinate.

### A.3 Proof of Theorem 3

For all $\eta \in \mathbb{C}^m$,

$$\mathbb{E}\left[\exp\left\{\langle \eta, \mathbf{z}_{t_2}\rangle\right\}\mid \mathbf{z}_{t_1}\right] = \frac{\sum_{u,v} \pi(u)Q(u,v)\gamma_u(\mathbf{z}_{t_1})\int \exp(\langle \eta, \mathbf{z}\rangle)\gamma_v(\mathbf{z})d\mathbf{z}}{\sum_u \pi(u)\gamma_u(\mathbf{z}_{t_1})}$$

$$= \frac{\sum_u \alpha_u(\eta)\pi(u)\gamma_u(\mathbf{z}_{t_1})}{\sum_u \pi(u)\gamma_u(\mathbf{z}_{t_1})},$$

with $\alpha_u(\eta) = \sum_v Q(u,v)\int \exp(\langle \eta, \mathbf{z}\rangle)\gamma_v(\mathbf{z})d\mathbf{z}$.

Assume that the emission densities $(\gamma_u)_{1\leqslant u\leqslant K}$ are linearly independent and $\pi(u) > 0$ for all $u \in \{1,\ldots,K\}$, then the only situation where $\mathbb{E}[\exp\{\langle \eta, \mathbf{z}_{t_2}\rangle\}|\mathbf{z}_{t_1}]$ is the null random variable is when $\alpha_u(\eta) = 0$ for all $u \in \{1,\ldots,K\}$. If the functions $(\eta \mapsto \int \exp(\langle \eta, \mathbf{z}\rangle)\gamma_v(\mathbf{z})d\mathbf{z})_{1\leqslant v\leqslant K}$ do not have simultaneous zeros and $Q$ has full rank, this is not possible.

### A.4 Proof of Theorem 4

We prove that the result holds for all $i = 1,\ldots,N$ and drop the index $i$ in this proof for ease of notation. Denote by

$$\Lambda := \begin{pmatrix} 1-p & p \\ q & 1-q \end{pmatrix}$$

the transition matrix of the hidden chain. Then, the stationary distribution is given by $\pi(0) = q/(p+q)$, $\pi(1) = p/(p+q)$, and the distribution of 2 consecutive observations is given by, for all $(a,b)$ in the support:

$$p_2(a,b) = \frac{q(1-p)}{p+q}\gamma_0(a)\gamma_0(b) + \frac{qp}{p+q}\gamma_0(a)\gamma_1(b) + \frac{pq}{p+q}\gamma_1(a)\gamma_0(b) + \frac{p(1-q)}{p+q}\gamma_1(a)\gamma_1(b).$$

If $Q_2 = \log p_2$ then simple computations lead to

$$(p+q)^2 p_2(a,b)^2 \frac{\partial^2 Q_2}{\partial a\partial b} = pq(1-p-q)(\gamma_0(a)\gamma_1'(a) - \gamma_0'(a)\gamma_1(a))(\gamma_0(b)\gamma_1'(b) - \gamma_0'(b)\gamma_1(b)).$$

Since $\gamma_0(a)\gamma_1'(a) - \gamma_0'(a)\gamma_1(a) = 0$ for $a$ in an open subset of the support if and only if on this interval $\gamma_0$ and $\gamma_1$ are proportional, assumption (B1) is satisfied if and only if on any open interval $\gamma_0^{(i)}$ and $\gamma_1^{(i)}$ are not proportional. Moreover, on the set of couples $(a,b)$ such that $\frac{\partial^2 Q_2}{\partial a\partial b} \neq 0$,

$$\log\left(\frac{\partial^2 Q_2}{\partial a\partial b}\right) = \log[|pq(1-p-q)|] - 2\log(p+q) - 2\log p_2(a,b) + h(a) + h(b),$$

where $h(a) = |\gamma_0(a)\gamma_1'(a) - \gamma_0'(a)\gamma_1(a)|$. We deduce easily that (B2) is satisfied if and only if on any open interval $\gamma_0^{(i)}$ and $\gamma_1^{(i)}$ are not proportional.

## B  Identifiability in Gaussian case

Theorem 2 has a condition on "non-quasi-Gaussianity" which is a generalization of the property of non-Gaussianity typical in ICA. Here, we consider the case of Gaussian noise-free data. Separation is actually possible by the temporal dependencies, but under a stricter condition. We put together results by Hyvärinen and Morioka (2017) and Belouchrani et al. (1997), and arrive at the following result:

**Theorem 5** *Assume the data follows the noise-free mixing model* $\mathbf{x}_t = \mathbf{f}(\mathbf{s}_t)$ *where* $\mathbf{s}_t$ *is a Gaussian process with independent components, and* $\mathbf{f}$ *is a* $\mathcal{C}^2$ *diffeomorphism with* $M = N$. *Assume further that*

- *The autocovariance functions* $c_i(\tau) = cov(s_t^{(i)}, s_{t-\tau}^{(i)})$ *are all distinct (i.e. any two of them for* $i, i'$ *are not equal). (Here,* $\tau$ *takes values in the set allowed by the definition of the index set.)*

*Then,* $\mathbf{f}^{-1}$ *and* $\mathbf{f}$ *can be recovered up to permutation and coordinate-wise linear transformations (applied on the components* $\mathbf{s}_t^{(i)}$*) from the distribution of* $\mathbf{x}_t$.

The proof is a straightforward implication of two theorems proven earlier: The nonlinear part is identifiable according to Theorem 2 by Hyvärinen and Morioka (2017) but a linear indeterminacy remains; here we need to note that $\bar{\alpha}$ in (Hyvärinen and Morioka, 2017) is a linear function for a Gaussian process. Subsequently the linear part can be identified, thanks to the autocovariance assumption above, as in Theorem 2 of Belouchrani et al. (1997).

Note that in the Gaussian case, it is not possible to apply Theorem 1 since (A3) cannot hold. Thus, Theorem 5 only applies for noise-free data.

## C   Learning and inference for $\Delta$-SNICA

The $\Delta$-SNICA generative model, as introduced in Section 3.2 can be written as:

$$p(u_1^{(i)}) = \prod_{k=1}^{K} (\pi_k^{(i)})^{\delta(u_1^{(i)}=k)} \tag{17}$$

$$p(u_t^{(i)} \mid u_{t-1}^{(i)}) = \prod_{k=1}^{K} \prod_{\ell=1}^{K} (A_{k\ell}^{(i)})^{\delta(u_t^{(i)}=k)\delta(u_{t-1}^{(i)}=\ell)} \tag{18}$$

$$p(\mathbf{y}_1^{(i)} \mid u_1^{(i)}) = \prod_{k=1}^{K} \mathcal{N}(\mathbf{y}_1^{(i)}; \bar{\mathbf{b}}_k^{(i)}, \bar{\mathbf{Q}}_k^{(i)})^{\delta(u_1^{(i)}=k)} \tag{19}$$

$$p(\mathbf{y}_t^{(i)} \mid \mathbf{y}_{t-1}^{(i)}, u_t^{(i)}) = \prod_{k=1}^{K} \mathcal{N}(\mathbf{y}_t^{(i)}; \mathbf{B}_k^{(i)}\mathbf{y}_{t-1}^{(i)} + \mathbf{b}_k^{(i)}, \mathbf{Q}_k^{(i)})^{\delta(u_t^{(i)}=k)} \tag{20}$$

$$p(\mathbf{x}_t \mid \mathbf{s}_t) = \mathcal{N}(\mathbf{x}_t; \mathbf{f}(\mathbf{s}_t), \mathbf{R}) \tag{21}$$

where the superscript $(i)$ again denotes that each independent component $i \in \{1, \ldots, N\}$ follows its own switching linear dynamical system. Also, as explained in Section 3.2, each independent component is part of a higher dimensional latent component $\mathbf{y}_t^{(i)} = (s_t^{(i)}, y_{t,2}^{(i)}, \ldots, y_{t,d}^{(i)})$. The mixing function $\mathbf{f}$ and other variables are defined as in the main text. The log-joint $\log \mathcal{L} = \log p(\mathbf{x}_{1:T}^{(1:N)}, \mathbf{y}_{1:T}^{(1:N)}, u_{1:T}^{(1:N)})$ can be written as:

$$\log \mathcal{L} = \sum_{t=1}^{T} \log p(\mathbf{x}_t \mid \mathbf{s}_t) + \sum_{i=1}^{N} \left( \log p(u_1^{(i)}) + \log p(\mathbf{y}_1^{(i)} \mid u_1^{(i)}) \right.$$
$$\left. \sum_{t=2}^{T} \log p(u_t^{(i)} \mid u_{t-1}^{(i)}) + \log p(\mathbf{y}_t^{(i)} \mid \mathbf{y}_{t-1}^{(i)}, u_t^{(i)}) \right). \tag{22}$$

The marginal likelihood is intractable and hence we instead optimize the variational evidence lower bound (ELBO), denoted here $\log \widehat{\mathcal{L}}$, under the assumption that the posterior factorizes as per

$$q(\mathbf{y}_{1:T}^{(1:N)}, u_{1:T}^{(1:N)}) = \prod_{i=1}^{N} q(\mathbf{y}_{1:T}^{(i)}) q(u_{1:T}^{(i)}). \tag{23}$$

The ELBO can thus be written as:

$$
\begin{aligned}
\log \widehat{\mathcal{L}} &= \mathbb{E}_q \left[ \log \frac{p(\mathbf{x}_{1:T}, \mathbf{y}_{1:T}^{(1:N)}, u_{1:T}^{(1:N)})}{q(\mathbf{y}_{1:T}^{(1:N)}, u_{1:T}^{(1:N)})} \right] \\
&= \mathbb{E}_q \left[ \sum_{t=1}^{T} \log p(\mathbf{x}_t \mid \mathbf{s}_t^{(1)}, ..., \mathbf{s}_t^{(N)}) + \sum_{i=1}^{N} \log \frac{p(\mathbf{y}_{1:T}^{(i)} \mid u_{1:T}^{(i)}) p(u_{1:T}^{(i)})}{q(\mathbf{y}_{1:T}^{(i)}) q(u_{1:T}^{(i)})} \right] \\
&= \mathbb{E}_q \left[ \sum_{t=1}^{T} \log p(\mathbf{x}_t \mid \mathbf{s}_t^{(1)}, ..., \mathbf{s}_t^{(N)}) \right] + \sum_{i=1}^{N} \left( - \mathrm{KL}\left[ q(u_{1:T}^{(i)}) \Big| p(u_{1:T}^{(i)}) \right] + \mathrm{H}\left[ q(\mathbf{y}_{1:T}^{(i)}) \right] \right. \\
&\quad \left. + \mathbb{E}_q \left[ \log p(\mathbf{y}_{1:T}^{(i)} \mid u_{1:T}^{(i)}) \right] \right) \\
&= \mathbb{E}_q \left[ \sum_{t=1}^{T} \log p(\mathbf{x}_t \mid \mathbf{s}_t^{(1)}, ..., \mathbf{s}_t^{(N)}) \right] + \sum_{i=1}^{N} \left( - \mathrm{KL}\left[ q(u_{1:T}^{(i)}) \Big| p(u_{1:T}^{(i)}) \right] + \mathrm{H}\left[ q(\mathbf{s}_{1:T}^{(i)}) \right] \right. \\
&\quad \left. + \mathbb{E}_q \left[ \log p(\mathbf{s}_1^{(i)} \mid u_1^{(i)}) \right] + \sum_{t=2}^{T} \mathbb{E}_q \left[ \log p(\mathbf{s}_t^{(i)} \mid \mathbf{s}_{t-1}^{(i)}, u_t^{(i)}) \right] \right) \tag{24}
\end{aligned}
$$

where H denotes Gaussian differential entropy, and $q$ is always with respect to the relevant variables. As long as all the distributions are conjugate-exponential families, we can use the Structured VAE Johnson et al. (2016) framework for inference and learning. We provide further detail on these two steps below.

**Inference**  Notice that we can write the latent variable part of our generative model in the following useful exponential family forms:

$$
\begin{aligned}
p(u_1^{(i)}) &= \prod_{k=1}^{K} \pi_k^{(i)\delta(u_1^{(i)}=k)} = \exp\left\{ \sum_{k=1}^{K} \delta(u_1^{(i)} = k) \log \pi_k^{(i)} \right\} = \exp\left\{ \langle \boldsymbol{\eta}_{\boldsymbol{\pi}}^{(i)}, \boldsymbol{\delta}_{u_1}^{(i)} \rangle \right\} \\
p(u_t^{(i)} \mid u_{t-1}^{(i)}) &= \prod_{k=1}^{K} \prod_{\ell=1}^{K} A_{k\ell}^{(i)\delta(u_{t-1}^{(i)}=k)\delta(u_t^{(i)}=\ell)} = \exp\left\{ \langle \boldsymbol{\eta}_{\mathbf{A}}^{(i)}, \boldsymbol{\delta}_{u_{t-1}, u_t}^{(i)} \rangle \right\} \tag{25} \\
p(\mathbf{y}_1^{(i)} \mid u_1^{(i)}) &= \prod_{k=1}^{K} \mathcal{N}(\mathbf{y}_1^{(i)}; \bar{\mathbf{b}}_k^{(i)}, \bar{\mathbf{Q}}_k^{-1(i)})^{\delta(u_1^{(i)}=k)} \\
&= \exp\left\{ \sum_{k=1}^{K} \delta(u_1^{(i)} = k) \left( \langle \mathbf{h}_{1,k}^{(i)}, \mathbf{y}_1^{(i)} \rangle + \mathbf{y}_1^{(i)^T} \mathbf{J}_{1,k}^{(i)} \mathbf{y}_1^{(i)} - \log Z_{1,k}^{(i)} \right) \right\} \\
\mathbf{h}_{1,k}^{(i)} &= \bar{\mathbf{Q}}_k^{(i)} \bar{\mathbf{b}}_k^{(i)} \\
\mathbf{J}_{1,k}^{(i)} &= -\frac{1}{2} \bar{\mathbf{Q}}_k^{(i)},
\end{aligned}
$$

where $\log Z_{1,k}^{(i)}$ is the log-normalizer, and similarly

$$p(\mathbf{y}_t^{(i)} \mid \mathbf{y}_{t-1}^{(i)}, u_t^{(i)}) = \prod_{k=1}^{K} \mathcal{N}(\mathbf{y}_t^{(i)}; \mathbf{B}_k^{(i)} \mathbf{y}_{t-1}^{(i)} + \mathbf{b}_k^{(i)}, \mathbf{Q}_k^{-1^{(i)}})^{\delta(u_t^{(i)} = k)}$$

$$= \exp \left\{ \sum_{k=1}^{K} \delta(u_t^{(i)} = k) \left( \left\langle \mathbf{h}_k^{(i)}, \mathbf{y}_{t-1,t}^{(i)} \right\rangle + \mathbf{y}_{t-1,t}^{(i)^T} \mathbf{J}_k^{(i)} \mathbf{y}_{t-1,t}^{(i)} - \log Z_k^{(i)} \right) \right\}$$

$$\mathbf{y}_{t-1,t}^{(i)} = (\mathbf{y}_{t-1}^{(i)}, \mathbf{y}_t^{(i)})^T$$

$$\mathbf{h}_k^{(i)} = \begin{pmatrix} \mathbf{B}_k^{(i)^T} \mathbf{Q}_k^{(i)} \mathbf{B}_k^{(i)} & -\mathbf{B}_k^{(i)^T} \mathbf{Q}_k^{(i)} \\ -\mathbf{Q}_k^{(i)} \mathbf{B}_k^{(i)} & \mathbf{Q}_k^{(i)} \end{pmatrix} \begin{pmatrix} \mathbf{0} \\ \mathbf{b}_k^{(i)} \end{pmatrix}$$

$$\mathbf{J}_k^{(i)} = -\frac{1}{2} \begin{pmatrix} \mathbf{B}_k^{(i)^T} \mathbf{Q}_k^{(i)} \mathbf{B}_k^{(i)} & -\mathbf{B}_k^{(i)^T} \mathbf{Q}_k^{(i)} \\ -\mathbf{Q}_k^{(i)} \mathbf{B}_k^{(i)} & \mathbf{Q}_k^{(i)} \end{pmatrix} .$$

Applying standard results from structured mean-field inference, the updates for the approximate posterior of the HMM latent variables is as follows:

$$q(u_{1:T}^{(i)}) \propto \exp \left\{ \log p(u_1^{(i)}) + \sum_{t=2}^{T} \log p(u_t^{(i)} \mid u_{t-1}^{(i)}) \right.$$

$$\left. + \mathbb{E}_{q(\mathbf{y}_1^{(i)})} \left[ \log p(\mathbf{y}_1^{(i)} \mid u_1^{(i)}) \right] + \mathbb{E}_{q(\mathbf{y}_{t-1,t}^{(i)})} \left[ \log p(\mathbf{y}_t^{(i)} \mid \mathbf{y}_{t-1}^{(i)}, u_t^{(i)}) \right] \right\} .$$

And by plugging in the distributions explicitly gives

$$q(u_{1:T}^{(i)}) \propto \exp \left\{ \langle \boldsymbol{\eta}_{\boldsymbol{\pi}^{(i)}}, \boldsymbol{\delta}_{u_1}^{(i)} \rangle + \langle \boldsymbol{\delta}_{u_1}^{(i)}, \boldsymbol{\rho}_1^{(i)} \rangle + \sum_{t=2}^{T} \langle \boldsymbol{\eta}_{\mathbf{A}^{(i)}}, \text{vec} \left( \boldsymbol{\delta}_{u_{t-1}}^{(i)} \boldsymbol{\delta}_{u_t}^{(i)^T} \right) \rangle + \langle \boldsymbol{\delta}_{u_t}^{(i)}, \boldsymbol{\rho}_t^{(i)} \rangle \right\} , \tag{26}$$

where we have defined

$$\mathbb{E}_{q(\mathbf{y}_{t-1,t}^{(i)})} \left[ \log p(\mathbf{y}_t^{(i)} \mid \mathbf{y}_{t-1}^{(i)}, u_t^{(i)}) \right] = \sum_{k=1}^{K} \delta(u_t^{(i)} = k) \, \mathbb{E}_{q(\mathbf{y}_{t-1,t}^{(i)})} \left[ \left\langle \mathbf{h}_{t,k}^{(i)}, \mathbf{y}_{t-1,t}^{(i)} \right\rangle + \right.$$

$$\left. \mathbf{y}_{t-1,t}^{(i)^T} \mathbf{J}_{t,k}^{(i)} \mathbf{y}_{t-1,t}^{(i)} - \log Z_{t,k}^{(i)} \right]$$

$$= \langle \boldsymbol{\delta}_{u_t}^{(i)}, \boldsymbol{\rho}_t^{(i)} \rangle .$$

Equation (26) can be viewed as a factor graph of unnormalized potentials – we can therefore use standard message passing algorithms for efficient inference. For instance, the forward-pass is:

$$\alpha(u_t^{(i)}) = \sum_{u_{t-1}} \exp \left\{ \sum_{t=2}^{T} \langle \boldsymbol{\eta}_{\mathbf{A}^{(i)}}, \text{vec} \left( \boldsymbol{\delta}_{u_{t-1}}^{(i)} \boldsymbol{\delta}_{u_t}^{(i)^T} \right) \rangle + \langle \boldsymbol{\delta}_{u_t}^{(i)}, \boldsymbol{\rho}_t^{(i)} \rangle \right\} \alpha(u_{t-1}^{(i)}) . \tag{27}$$

Similarly, the standard mean-field updates for the dynamical system latent variables gives:

$$q(\mathbf{y}_{1:T}^{(i)}) \propto \exp \left\{ \sum_{t=1}^{T} \mathbb{E}_{\prod_{j=1}^{N \setminus i} q(\mathbf{y}_t^{(j)})} \left[ \log p(\mathbf{x}_t \mid \mathbf{s}_t) \right] + \mathbb{E}_{q(u_1^{(i)})} \left[ \log p(\mathbf{y}_1^{(i)} \mid u_1^{(i)}) \right] \right.$$

$$\left. + \sum_{t=2}^{T} \mathbb{E}_{q(u_t^{(i)})} \left[ \log p(\mathbf{y}_t^{(i)} \mid \mathbf{y}_{t-1}^{(i)}, u_t^{(i)}) \right] \right\} . \tag{28}$$

The problem here is that we would like to write all the factors in terms of $\mathbf{s}_t$ and $\mathbf{y}_t$ conditonal on $\mathbf{x}_t$. However, due to the nonlinear mixing function, we can't write this directly in conjugate exponential family form. To resolve this, we follow Johnson et al. (2016) and use a decoder neural network to predict approximate natural parameters such that they are in conjugate form, namely:

$$\mathbb{E}_{\prod_{N \setminus i} q(\mathbf{y}_t^{(j)})} \left[ \log p(\mathbf{x}_t \mid \mathbf{s}_t) \right] \propto \langle \mathbf{v}_t(\mathbf{x}_t; \boldsymbol{\phi}), \mathbf{s}_t \rangle + \mathbf{s}_t^T \mathbf{W}_t(\mathbf{x}_t; \boldsymbol{\phi}) \mathbf{s}_t ,$$

where $\mathbf{v}_t, \mathbf{W}_t$ are thus the outputs of the decoder network, with the latter term assumed to have diagonal structure with negative entries to ensure it's an appropriate Gaussian natural parameter. Further, due to the factored approximation assumption over $\mathbf{y}_t^{(1)}, \ldots, \mathbf{y}_t^{(N)}$ and thus $\mathbf{s}_t^{(1)}, \ldots, \mathbf{s}_t^{(N)}$, above can be written as:

$$
\mathbb{E}_{\prod_{N \setminus i} q(\mathbf{y}_t^{(j)})} \left[ \log p(\mathbf{x}_t \mid \mathbf{s}_t) \right] \propto \left( v_{t,i} + 2 \sum_{j \setminus i}^N w_{t,j,i} \, \mathbb{E}_{q(\mathbf{y}_t^{(j)})} \left[ y_{t,1}^{(j)} \right] \right) y_{t,1}^{(i)} + w_{t,i,i} y_{t,1}^{(i)^2}
$$

$$
= \langle \tilde{\mathbf{v}}_t^{(i)}, \mathbf{y}_t^{(i)} \rangle + \mathbf{y}_t^{(i)^T} \widetilde{\mathbf{W}}^{(i)} \mathbf{y}_t^{(i)} \tag{29}
$$

where $\tilde{\mathbf{v}}_t^{(i)}, \widetilde{\mathbf{W}}^{(i)}$ are zero everywhere except in their first indices. The other expectations in Equation (28) are just responsibility weighted natural parameters. For instance:

$$
\mathbb{E}_{q(u_t^{(i)})} \left[ \log p(\mathbf{y}_t^{(i)} \mid \mathbf{y}_{t-1}^{(i)}, u_t^{(i)}) \right] \propto \sum_{k=1}^K \mathbb{E}_{q(u_t^{(i)})} \left[ \delta(u_t^{(i)} = k) \right] \left( \langle \mathbf{h}_{t,k}^{(i)}, \mathbf{y}_{t-1,t}^{(i)} \rangle + \mathbf{y}_{t-1,t}^{(i)^T} \mathbf{J}_{t,k}^{(i)} \mathbf{y}_{t-1,t}^{(i)} \right)
$$

$$
\propto \left\langle \tilde{\mathbf{h}}_t^{(i)}, \mathbf{y}_{t-1,t}^{(i)} \right\rangle + \mathbf{y}_{t-1,t}^{(i)^T} \tilde{\mathbf{J}}_t^{(i)} \mathbf{y}_{t-1,t}^{(i)}
$$

$$
\tilde{\mathbf{h}}_t^{(i)} = \sum_{k=1}^K \mathbb{E}_{q(u_t^{(i)})} \left[ \delta(u_t^{(i)} = k) \right] \mathbf{h}_{t,k}^{(i)}
$$

$$
\tilde{\mathbf{J}}_t^{(i)} = \sum_{k=1}^K \mathbb{E}_{q(u_t^{(i)})} \left[ \delta(u_t^{(i)} = k) \right] \mathbf{J}_{t,k}^{(i)}
$$

The approximate posterior in (28) can therefore be written as:

$$
q(\mathbf{y}_{1:T}^{(i)}) \propto \exp \Big\{ \langle \tilde{\mathbf{v}}_1^{(i)}, \mathbf{y}_1^{(i)} \rangle + \mathbf{y}_1^{(i)^T} \widetilde{\mathbf{W}}^{(i)} \mathbf{y}_1^{(i)} + \langle \tilde{\mathbf{h}}_1^{(i)}, \mathbf{y}_1^{(i)} \rangle + \mathbf{y}_1^{(i)^T} \tilde{\mathbf{J}}_1^{(i)} \mathbf{y}_1^{(i)}
$$

$$
+ \sum_{t=2}^T \langle \tilde{\mathbf{v}}_t^{(i)}, \mathbf{y}_t^{(i)} \rangle + \mathbf{y}_t^{(i)^T} \widetilde{\mathbf{W}}^{(i)} \mathbf{y}_t^{(i)} + \left\langle \tilde{\mathbf{h}}_t^{(i)}, \mathbf{y}_{t-1,t}^{(i)} \right\rangle + \mathbf{y}_{t-1,t}^{(i)^T} \tilde{\mathbf{J}}_t^{(i)} \mathbf{y}_{t-1,t}^{(i)} \Big\}. \tag{30}
$$

This can again be viewed as a factor graph on which to perform message passing. The initial forward message is

$$
\alpha(\mathbf{y}_1) = \exp \left\{ \langle \tilde{\mathbf{v}}_1 + \tilde{\mathbf{h}}_1, \mathbf{y}_1 \rangle + \mathbf{y}_1^T \left( \widetilde{\mathbf{W}} + \tilde{\mathbf{J}}_1 \right) \mathbf{y}_1 \right\},
$$

$$
= \exp \left\{ \langle \boldsymbol{\eta}_1, \mathbf{y}_1 \rangle + \mathbf{y}_1^T \mathbf{P}_1 \mathbf{y}_1 \right\},
$$

which is an unnormalized Gaussian distribution, and we have dropped superscripts for convenience. The forward equations can be derived as follows, shown here for $t - 1 = 1, t = 2$:

$$
\alpha(\mathbf{y}_2) = \exp\{ \langle \tilde{\mathbf{v}}_2, \mathbf{y}_2 \rangle + \mathbf{y}_2^T \widetilde{\mathbf{W}} \mathbf{y}_2 \} \int_{\mathbf{y}_1} \exp \left\{ \left\langle \tilde{\mathbf{h}}_2, \mathbf{y}_{1,2} \right\rangle + \mathbf{y}_{1,2}^T \tilde{\mathbf{J}}_2 \mathbf{y}_{1,2} + \langle \boldsymbol{\eta}_1, \mathbf{y}_1 \rangle + \mathbf{y}_1^T \mathbf{P}_1 \mathbf{y}_1 \right\}.
$$

Define $\boldsymbol{\eta}_2^* = (\tilde{\mathbf{h}}_2^1 + \boldsymbol{\eta}_1, \tilde{\mathbf{h}}_2^2)^T$ and $\mathbf{P}_2^* = \begin{pmatrix} \tilde{\mathbf{J}}_2^{11} + \mathbf{P}_1 & \tilde{\mathbf{J}}_2^{12} \\ \tilde{\mathbf{J}}_2^{21} & \tilde{\mathbf{J}}_2^{22} \end{pmatrix}$ with the superscripts denoting block partitions corresponding to $\mathbf{y}_1$ and $\mathbf{y}_2$, so that

$$
\alpha(\mathbf{y}_2) = \exp\{ \langle \tilde{\mathbf{v}}_2, \mathbf{y}_2 \rangle + \mathbf{y}_2^T \widetilde{\mathbf{W}} \mathbf{y}_2 \} \int_{\mathbf{y}_1} \exp \left\{ \langle \boldsymbol{\eta}_2^*, \mathbf{y}_{1,2} \rangle + \mathbf{y}_{1,2}^T \mathbf{P}_2^* \mathbf{y}_{1,2} \right\},
$$

where the integral is (unnormalized) joint Gaussian on $(\mathbf{y}_1, \mathbf{y}_2)^T$ with $\boldsymbol{\mu} = -\frac{1}{2} \mathbf{P}_2^{*-1} \boldsymbol{\eta}_2^*$ and $\boldsymbol{\Lambda} = -2 \mathbf{P}_2^*$. The block marginalization properties of Gaussian distributions gives:

$$
\alpha(\mathbf{y}_2) = \exp\{ \langle \tilde{\mathbf{v}}_2, \mathbf{y}_2 \rangle + \mathbf{y}_2^T \widetilde{\mathbf{W}} \mathbf{y}_2 \} \exp \left\{ \langle \boldsymbol{\eta}_2, \mathbf{y}_2 \rangle + \mathbf{y}_2^T \mathbf{P}_2 \mathbf{y}_2 \right\},
$$

with

$$
\boldsymbol{\eta}_2 = \tilde{\mathbf{h}}_2^2 - \tilde{\mathbf{J}}_2^{21} (\tilde{\mathbf{J}}_2^{11} + \mathbf{P}_1)^{-1} (\tilde{\mathbf{h}}_2^1 + \boldsymbol{\eta}_1)
$$

$$
\mathbf{P}_2 = \tilde{\mathbf{J}}_2^{22} - \tilde{\mathbf{J}}_2^{21} (\tilde{\mathbf{J}}_2^{11} + \mathbf{P}_1)^{-1} \tilde{\mathbf{J}}_2^{12}
$$

Thus, the message passing on the linear dynamical system ends up as updates on the natural parameters:

$$\alpha(\mathbf{y}_2) = \exp\left\{\langle \tilde{\mathbf{v}}_2 + \boldsymbol{\eta}_2, \mathbf{y}_2 \rangle + \mathbf{y}_2^T \left(\widetilde{\mathbf{W}} + \mathbf{P}_2\right) \mathbf{y}_2\right\},$$

which is analogous to the Kalman filter updates. Similar update equations can be derived for the backward pass and the marginal posteriors are given by the normalized product of the forward and backward passes. Since the resulting distributions are Gaussian, it is easy to compute the expected sufficient statistics required in the inference step described above for $q(u_{1:T})$. In practice, we will cycle between these two inference steps until convergence, after which the M-step is carried out.

**Learning**   After repeating the inference step until convergence, we perform stochastic gradient updates by maximizing the ELBO (Equation (24)) with respect to all the model parameters. In particular, to optimize the first term:

$$\mathbb{E}_q\left[\sum_{t=1}^T \log p(\mathbf{x}_t \mid \mathbf{s}_t^{(1)}, ..., \mathbf{s}_t^{(N)})\right]$$

we sample $\mathbf{s}_t^{(1:N)} \sim q(\mathbf{s}_t^{(1:N)}), \forall t \in (1, \dots, T)$, and parameterize the mixing function with a decoder neural network $\mathbf{f}(\cdot; \boldsymbol{\theta})$:

$$p(\mathbf{x}_t \mid \mathbf{s}_t) = \mathcal{N}(\mathbf{x}_t; \mathbf{f}(\mathbf{s}_t; \boldsymbol{\theta}), \mathbf{R}).\tag{31}$$

# D   Details on experiments on simulated data

**Simulated data**   We simulated 100K long time-sequences from the $\Delta$-SNICA and computed the mean absolute correlation coefficient (MCC) between the estimated latent components and ground true independent components. The switching linear dynamical system was simulated to have two latent hmm states, one that induced strong mean reverting behaviour upon the linear dynamical system, and another with oscillatory dynamics. The dimension of the linear dynamical system state-space was also set to 2 (1 + independent component). The HMM transition matrix was close to diagonal with 0.99 probability of staying in current state and 0.01 probability of transitioning to the other state, at each time step of the 100k long sequence. The code at [redacted for anonymity] provides the exact simulation details. To illustrate the dimensionality reduction capabilities we considered two settings where the observed data dimension $M$, was either 12 or 24 and the number of independent components, $N$ was 3 and 6, respectively. Therefore the model consist of $N$ independent processes of Equation (4). Observations were created by the mixing function (Eq. (3)) and additive Gaussian diagonal noise. We considered four levels of mixing of increasing complexity by randomly initialized MLPs of the following number of layers: 1 (linear ICA), 2, 3, and 5.

**Training details**   All the experiments were run on ten different randomly simulated data sets to compute error bars. The model parameters, including the mixing function, were estimated using the inference and learning algorithm described above. All parameters were trained in ordered to increase the ELBO of the model; Adam with learning rate 1e-2 was used. The number of layers in the decoder networks was set equal to the number of mixing layers for both $\Delta$-SNICA and IIA-HMM benchmark. The number of layers in the encoder $\Delta$-SNICA was always one more than that for the decoder. We suspect this extra nonlinearity in the encoder helped training since VAEs have tendency to over-emphasize learning the likelihood term, which this may have alleviated. The number of hidden units was set at 128 and 64 for the decoder and encoder respectively. In order to avoid local minima, we started training from 20 different inital seeds and chose the model that reached the highest ELBO, or likelihood. The models were trained on University of Helsinki SLURM cluster until convergence, which in practice was approximately 12 hours on most settings. All training was done on CPUs only. Memory used for a single model to be trained was 15G RAM.

**Further discussion of results**   One possible reason for the relatively poor performance of IIA-HMM on the simulated data experiment (Figure 2) was suspected to be loss of information that resulted from the PCA preprocessing step. We explored this in additional experiments where there was no dimension reduction: $\Delta$-SNICA still outperforms IIA-HHM also in this setting, although the latter's performance is now improved for small dimensionality (in 3-dimensions: MCC avg.

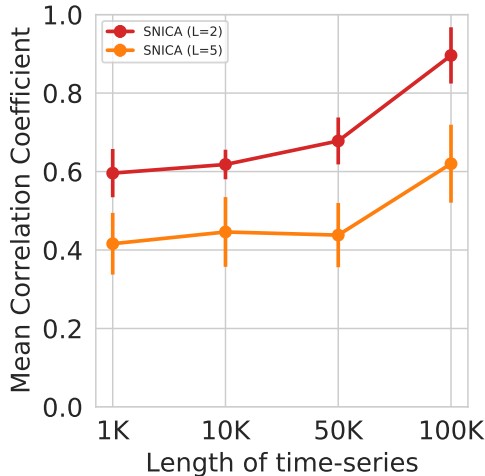

Figure 4: Mean absolute correlation coefficient between estimated and ground true independent components for varying lengths of training data for Δ-SNICA (N=3, M=12), for equal training time. Result shown for two different numbers of mixing layers L=2 and L=5

0.4 for 3 mixing layers), though remains clearly below Δ-SNICA (3-dimensions: MCC avg. 0.7). IIA-HMM performance for dimensions above 6, even without dimension reduction, was very poor (MCC < 0.3) suggesting its poor performance is not solely due to PCA, bur rather likely due to it lacking observation noise model (unlike all the other models we considered) and simpler model of latent dynamics (original iVAE also has no latent dynamics model but was here supplied with the ground-truth HMM latent state thus giving it a substantial advantage over IIA-HMM).

**Size of training data** The theoretical identifiability results presented in this paper hold in the limit of infinite data. Hence, we hypothesized that the amount of training data may have large impact in any practical situations – in addition to the usual benefits of increased dataset size. To explore this, we trained our model for varying lengths of datasets, with the results shown in Figure 4. We observed much better results for the largest dataset. Due to limited compute available to us, we leave it for future works to investigate even larger data sizes.

## E    Details on MEG experiment

**Data and Preprocessing** The MEG data used were from the open Cam-CAN data repository[3] (available at http://www.mrc-cbu.cam.ac.uk/datasets/camcan/), and released under Creative Commons license. (Taylor et al., 2017; Shafto et al., 2014). The MEG dataset was collected using a 306-channel VectorView MEG system (Elekta Neuromag, Helsinki), consisting of 102 magnetometers and 204 orthogonal planar gradiometers with sampling 1000Hz. MEG data was preprocessed by temporal signal space separation (tsss; MaxFilter 2.2, Elekta Neuromag Oy, Helsinki, Finland) to remove noise from external sources and from HPI coils and head-motion was corrected (see (Taylor et al., 2017) for more details of the preprocessing). During the resting state recording, subjects sat still with their eyes closed for at least 8 min and 40 s. In the task-session data, the subjects carried out a (passive) audio–visual task including 120 trials of unimodal stimuli (60 visual stimuli: bilateral/full-field circular checkerboards; 60 auditory stimuli: binaural tones), presented at a rate of approximately 1 per second. In this study, We applied the method to 10 subjects' data and downsampled it to 128 Hz for saving computational resources, and only data from the planar gradiometers (204 channels) were used. We further band-pass filtered the data between 4 Hz and 30 Hz and normalized them to have

---

[3]Acknowledgment for Cam-CAN data: Data collection and sharing for this project was provided by the Cambridge Centre for Ageing and Neuroscience (CamCAN). CamCAN funding was provided by the UK Biotechnology and Biological Sciences Research Council (grant number BB/H008217/1), together with support from the UK Medical Research Council and University of Cambridge, UK.

zero-mean and unit variance. For the task-session data, we cropped each trial from -300ms to 600ms after the onset. The MNE package (Gramfort et al., 2013) was used for preprocessing.

**SNICA setting**  We only used resting-state data for training. For saving memory, we selected 5-min long resting-state data from each subject. We temporally concatenated segments of each subject to form a dataset (5*60*128*10 = 384k time points) for training. We fixed the number of independent components to 5, and set the number of hidden markov states and the dimension of the linear dynamical system to 2. The number of layesr in the encoder and decoder networks was set equal, and the number of hidden units was set to 32. Otherwise, all the settings were as in Simulation.

**Evaluation Methods**  For evaluation, we used the model trained with (unlabeled) resting-state data as feature extractors to perform a downstream task for classification of (labeled) task-session data. We carried out classification of the stimulus modality (auditory or visual) by using the estimated features. Classification was performed using a linear support vector machine (SVM) classifier trained on the stimulation modality labels and sliding-window-averaged features (width=10 and stride=3 samples) for each trial. The performance was evaluated by the generalizability of a classifier across subjects, i.e., one-subject-out cross-validation (OSO-CV). The hyperparameters of the SVM were determined by nested OSO-CV without using the test data. For comparison, IIA-HMM and IIA-TCL for the nonlinear vector autoregressive model (NVAM) were applied as baseline methods. Since IIA-HMM is not able to reduce the dimensionality, PCA was performed on the concatenated resting-state data to reduce the dimension to 5 for fair comparison. For IIA-TCL, we used segments of equal size, of length 10 s or 1280 data points, and also set the number of independent innovation to 5 for fair comparison.

We visualized the spatial patterns of the estimated features by plotting the weight vectors of units from encoder MLP in the topography map space. For the first layer, we have weight vectors (columns of the weight matrix $\mathbf{W}_1$) across sensors for each unit, and directly mapped them into brain topography space. And the weight matrix $\mathbf{W}_2$ multiplied by $\mathbf{W}_1$ to obtain weight vectors (columns of $\mathbf{W}_1\mathbf{W}_2$) of sensors for each unit in the second layer, and so on for subsequent layers.

**Interpreting the latent dynamics in the MEG experiments**  The learned parameters for the $\Delta$-SNICA 's latent dynamics, namely HMM-style switching, provide interesting interpretations in the MEG data experiment. Since we fix the number of HMM states to be two for each component, our assumption is that they can be interpreted as on/off or activity/inactivity. Such long-term on/off switching of the sources thus characterizes the nonstationary of the brain signal, as is quite often assumed in brain imaging. In particular, the components can be interpreted to represent different dynamic brain processes that are well-known to exist in the resting brain: visual, auditory, and other sensory networks; executive networks, attentional networks, and default mode network. The specific transition matrices for the hidden Markov discrete states can be interpreted to represent the movement between the transient brain states (process) in the real data. In particular, we found the HMM transition matrix to be close to diagonal, which suggests that we are capturing relatively slowly evolving states. The precise figures from the transition matrix suggest that on average a given state (active/inactive) lasts between 0.8 and 7 seconds. The marginal probabilities of the different states are fairly similar to each other, ranging between 0.3 to 0.6, thus all the states are relatively common in this sense. The hidden continuous states, on the other hand, are used here mainly as an algorithmic trick to easily model higher-order AR processes, and are thus harder to interpet.