# OpenReview forum: "Disentangling Identifiable Features from Noisy Data with Structured Nonlinear ICA"
_NeurIPS.cc/2021/Conference — NeurIPS 2021 Poster_

### Official Review · Reviewer_x1ZH · 2021-07-03

**Rating:** 7
**Confidence:** 3

**Summary:**

This paper proposed a new general identifiable nonlinear ICA framework for structured data. They proved identifiability for this general framework. They also proposed a specific application model from the framework and applied it to the simulated and real data.


**Limitations And Societal Impact:**

The authors have discussed the limitations in detail in the paper.

**Main Review:**

Originality: Yes, the framework they proposed is more general than the previous papers. The related work is properly cited.

Quality: The theory part seems technically sound. I have some questions about the experimental part.
1. The simulated data is from the true SNICA model. The bad performance of IIA-HMM can be due to the model mismatch or the loss of information from the PCA pre-processing. Have you tried eliminating those effects in the comparison?
2. How do you select the number of independent components to be 5 in the real data application?
3. In real data, PCA was performed before applying IIA-HMM to the data, while SNICA directly applying to the data. PCA can loss a lot information. Are the comparison still fair? What's the classification accuracy if directly using the features extracted by a 5d PCA to predict the labels?
4. What do the inferred discrete states, continuous states, and transition matrices look like in real data? Do they have any interpretable meanings?

Clarity: Yes, the paper is well written. But I do want clarification for: in line 343-345, you claimed that SNICA has identifiability with noise of unknown, arbitrary distribution, while (Khemakhem et al., 2020a) assumed noise of known distribution. If one specifies a wrong noise distribution in inference, can SNICA still recover the true latents? According to Theorem 1 in Khemakhem et al., 2020a, they only have assumptions on the characteristic function of the noise distribution. Can you clarify what do you mean by saying noise of known distribution?

Significance: Yes, the results are important. The framework is general, and many models can be builded upon it.

Overall, I think this paper is theoretically sound, yet the experiment can be further improved. The framework is general, which leaves room for possible future works.

**Time Spent Reviewing:**

3

---

> ### Author Response · Authors · 2021-08-10
> **Author response #1**
>
> Dear reviewer, we very much appreciate your kind comments and insightful questions. Below we provide some clarifications.
>
> **"The simulated data is from the true SNICA model. The bad performance of IIA-HMM can be due to the model mismatch or the loss of information from the PCA pre-processing. Have you tried eliminating those effects in the comparison?"** One of the problems with IIA-HMM that we wanted to improve upon is its inability to perform dimension reduction without PCA, or similar, and it was our purpose to highlight this on the simulated data -- in this sense we find our experiment to be appropriate. However, you are correct that PCA pre-processing doesn't allow direct comparison of the models themselves. For this reason we have now run a new experiments on simulated data where the observed and latent components are of equal dimension and thus no PCA is required. SNICA still outperforms IIA-HHM also in this setting, although the latter's performance is now improved for small dimensionality (3d: e.g. MCC avg. 0.4 for 3 mixing layers), though remains clearly below SNICA (3d: MCC avg. 0.7). IIA-HMM performance for dimensions above 6 is very poor (MCC < 0.3) suggesting it's poor performance is not solely due to PCA.  We will include this figure in our final version of the paper. Further, we are also running the experiments with iVAE as an additional benchmark: iVAE is able to perform dimension reduction but the reason we didn't include it initially is that it cannot handle latent variables, nor their temporal dependencies. To resolve this, we have allowed iVAE to "cheat" by supplying the model with the latent HMM states as its observed auxiliary variables. This gives us a very challenging benchmark. We are currently in the process of finalizing these simulations and will inform you of the results as soon as they finish.
>
> **"How do you select the number of independent components to be 5 in the real data application?"**
> Here, our goal is to compare with IIA-HMM so we set the dimension to be consistent with the paper proposing the IIA-HMM method for a fair comparison, where they used 5 dimensions on similar MEG data.
>
> **"In real data, PCA was performed before applying IIA-HMM to the data, while SNICA directly applying to the data. PCA can loss a lot information. Are the comparison still fair? What's the classification accuracy if directly using the features extracted by a 5d PCA to predict the labels?"**
> We would argue that this is a fair comparison since it is exactly the approach taken in the the original IIA-HMM paper. It also highlights the problem we try to address with our framework, namely that previous nonlinear ICA works are not able to perform dimension reduction on their own. Furthermore, the performance of IIA-HMM with PCA preprocessing is substantially better than just PCA, which also suggests that it's a good baseline. To quantify this, we have now run new experiment with classification by PCA only, as you requested, and the precise classification accuracy on the real data are 72\%, which is substantially lower than with the other methods (average 85-90\% for SNICA, and 80-83\% for IIA). We will include this new baseline in our figure for the final version of the paper.
>
> **"What do the inferred discrete states, continuous states, and transition matrices look like in real data? Do they have any interpretable meanings?"** Since we fix the number of states to be two for each component, our assumption is that they can be interpreted as on/off or activity/inactivity. Such long-term on/off switching of the sources thus characterizes the nonstationary of the brain signal, as is quite often assumed in brain imaging. In particular, the components can be interpreted to represent different dynamic brain processes that are well-known to exist in the resting brain: visual, auditory, and other sensory networks; executive networks, attentional networks, and last but no least, (parts of) default mode network. The transition matrices for discrete states can be interpreted to represent the movement between the transient brain states (process) in the real data. In particular, we found the HMM transition matrix to be close to diagonal, which suggests that we are capturing relatively slowly evolving states. The precise figures from the transition matrix suggest that on average a given state (active/inactive) lasts between 0.8 and 7 seconds. The marginal probabilities of the different states are fairly similar to each other, ranging between 0.3 to 0.6, thus all the states are relatively common in this sense. The hidden continuous states, on the other hand, are used here mainly as an algorithmic trick to easily model higher-order AR processes, so we would not interpret them in detail. We will add this discussin in our final manuscript.
>
> **"In line 343-345, you claimed that SNICA has identifiability with noise of unknown, arbitrary distribution, while (Khemakhem et al., 2020a) assumed noise of known distribution. If one specifies a wrong noise distribution in inference, can SNICA still recover the true latents? According to Theorem 1 in Khemakhem et al., 2020a, they only have assumptions on the characteristic function of the noise distribution. Can you clarify what do you mean by saying noise of known distribution?"**
> The meaning is that in SNICA we do not need to know the noise distribution, we can learn it. This is in contrast to Khemakhem et al (2020a), where the noise distribution is assumed to be fully known a priori, and cannot be learned, as is clear from their proof in Appendix B.2.2, equations (20) to (27). (Indeed, in that proof, both possible sets of distributions use the same noise distribution on each side of the equations, which could not be the case if the noise distribution was not assumed fully known. Then they have a supplementary assumption on the characteristic function to be able to go from (25) to (26).) But in either case we cannot use a wrong noise distribution, to have correct inference on the mixing system. So, if in our model we infer the noise distribution incorrectly (perhaps due to an inflexible noise model), or an incorrect noise model is assumed by Khemakhem et al, the learning goes wrong in the sense of not producing a consistent estimator. (At least, this is our thinking at the moment, although it is possible that some kind of robustness properties exist regarding specification of the noise distribution.)

---

> > ### Comment · Reviewer_x1ZH · 2021-08-25
> > **Response to authors' rebuttal**
> >
> > I thank the authors for their detailed rebuttal. They addressed my questions, and I'm willing to raise my score by 1 point.

---

### Official Review · Reviewer_yFEi · 2021-07-16

**Rating:** 7
**Confidence:** 4

**Summary:**

The authors present a new disentanglement framework for structured data. They claim that the method generalizes to scenarios with spatial dependencies, independent additive noise, and non-square mixing processes. They demonstrate the utility of the framework by applying an approximate model to toy data as well as MEG data.

**Limitations And Societal Impact:**

Yes, in the checklist response

**Main Review:**

*Review summary:*

As far as I know, SNICA is indeed a novel method for nonlinear ICA. The combined departure from typically requiring the latent and data dimensionality to be equal and the explicit noise term in equation 3 allow for a broader set of application spaces. My primary complaint is that the paper makes a number of strong claims about the novelty and superiority of the framework with insufficient evidence to support them. I would request that the authors put more effort into supporting their assertions, or dial them back, before the paper is ready for publication.

*Strengths:*

Overall the paper is well written. The mathematics in the main manuscript are thorough, complete, and easy enough to follow (I only skimmed the proofs in the appendix). The overall objective to push identifiable nonlinear ICA towards more practical regimes of noisy data and lower-dimensional latent structure is a valuable direction for the community. The application to real (MEG) data is also a plus, as many prior works in this space only test on simulated data.

*Weaknesses:*

I see two primary weaknesses in this paper: 1) numerous sweeping claims are made regarding their superiority over previously published results without sufficient support. 2) The empirical demonstration of the model compares against control models that are not well fit to the tasks. Both of these weaknesses are easily addressable, by 1) either supporting their claims more carefully or dialing them back and 2) choosing a more reasonable control or experimental task.

1) The authors make a large number of bold claims, for example in lines 9-10; 44-47; 52-53; 67; 96-99; 110; 123-125; 139; 141-142; 166; 300-301; 339-341. While I do not know the literature well enough to directly refute all of them, I will note a few examples where they appear to be wrong. I believe that this weakness can be resolved by either softening the claims themselves (which, in my opinion, would not reduce the novelty or significance of the submission) or providing a more detailed review of how previous works compare.

1. a) On lines 44-47 the authors note that “all models cited above, with the exception of Khemakhem et al. (2020a), assume that the data are fully observed and noise-free”. While that might be true, I immediately thought of the work of Locatello et al. {1}, which was not cited by the authors, but is an identifiable disentanglement method and comments on their ability to handle noise: “The generative model does not model additive noise; we assume that the noise is explicitly modeled as a latent variable and its effect is manifested through [the ground truth generator], as is done by [several citations].” (last paragraph of section 3) I admit that this might not necessarily be a contradictory claim, but I nonetheless would appreciate it if the authors clarified how their method of explicitly modeling additive noise is beneficial over the type of setup in {1} and the citations in Locatello et al.’s quoted sentence.

1.b) On lines 96-97 the authors note that “the mixing function f is assumed bijective and thus dimension reduction is not possible in most of the above models. The only exception is Khemakhem et al. (2020a) who…” I find the second part of this quote (“dimension reduction is not possible”) surprising in general, considering that any publication attempting to perform disentanglement on the DisentanglementLib dataset necessarily has to assume dimensionality reduction. I thought maybe the authors intended to say that the theory does not allow dimensionality reduction, even if the practical implementations do. However, Klindt et al., which is cited earlier, presents an identifiable disentanglement framework that only assumes an injective mixing function in their theory and therefore allows for dimensionality reduction.

1.c) The authors claim that “under the conditions given in the next section, we can now guarantee identifiability for a very broad and rich class of models. First, notice that all previous Nonlinear ICA time-series models can be recast and often improved upon when viewed through this new unifying framework.” I do not understand how this could possibly be true given that not all previous Nonlinear ICA work abides by the conditions given in sections 3 and 4. Just as two examples, the requirement for unconditional independence on lines 114-115, and tail behavior in assumption A1 are not ubiquitous in the literature. I could definitely be misunderstanding what the authors intended with their statement, in which case I would be happy with a clarification.

I understand that it is not reasonable for the authors to precisely state how every previous contribution fails to meet their claimed advantages. However, given the examples above, I found myself unconvinced in general. I further found myself wondering if it was necessary to make so many sweeping claims in the first place.

2) The authors set up a simulated example problem in section 5.1 to understand how their model works with a restricted experiment. I agree that this is an important step for understanding how the model behaves. They choose to compare against what they claim to be the state of the art, IIA-HMM. However, they state themselves that “IIA-HMM has a much simpler model of dynamics and no noise model, and likely lost information due to PCA pre-processing.” This leaves me wondering why they felt that this was a fair comparison, given that the problem setup is not at all matched to what IIA-HMM was designed to solve. I would be curious to see how they do for a non-dimensionality-reduced setup, since as far as I know their framework does not require the number of generating latents to be fewer than the data dimensionality. Or, a better solution would be for them to compare their model against an alternative that is appropriately matched to the dimensionality reduction task. Furthermore, the authors do not provide a baseline comparison for their denoising task. This might be because the authors wished to focus on identifiable models, which restricts the otherwise large set of denoising methods. However, again given the work cited in {1}, I am unconvinced that there was not a suitable comparison to be made. Without appropriate comparisons, we are left to rely solely on the scalar MCC metric in an unfamiliar simulated example, which I find insufficient.

*Additional minor complaints:*
1) The citations need to be revised and edited. Many are listed as arxiv prints that are now published in peer-reviewed venues. A few are missing the venue entirely. Morioka 2020b appears to be listed twice.

2) I noticed typos on line 243 (“a complete statistics”), fig 2 caption (“ground true independent”)

3) I think it would be helpful for practitioners if the authors included some justification for their decision on line 318, or at least a discussion on the tradeoffs for the number of independent components chosen.

4) I would appreciate it if the authors spent more time discussing trade-offs made in their framework. As of now it is limited to the first two sentences of the “Limitations” section starting on line 367. I found this to be rather terse given the space allocated for stating the failures of prior work. For example, if I understand the unconditional independence assumption on lines 114-115 correctly, then it seems highly unlikely to be met in real-world data, including their MEG experiment. The constraint on the distribution tails and 2nd order nature of the generator also seem restricting with respect to real data. Perhaps the authors could note alternative work that does not require such assumptions for identifiability (in exchange for other restrictions), which would assist future researchers seeking a more general solution.

{1} http://proceedings.mlr.press/v119/locatello20a.html

**Time Spent Reviewing:**

4

---

> ### Author Response · Authors · 2021-08-10
> **Author response #1**
>
> Dear reviewer, thank you very much for this most excellent and thorough review.
>
> Your first criticism was that **"numerous sweeping claims are made ..."**. In response to your comments, we will scale back and edit our text appropriately and provide stronger evidence where needed. Please see below for detailed comments on the lines that you pointed out. We will use **bold** to indicate reviewer comments, and *italics* for quotations from our original manuscript.:
>
> lines 9-10: We will delete this sentence as rest of the abstract already covers this.
>
> lines 44-47:*'all models cited above, with the exception of Khemakhem et al. (2020a), assume that the data are fully observed and noise-free'* **"1. a) While that might be true, I immediately thought of the work of Locatello et al. {1}, which was not cited by the authors, but is an identifiable disentanglement method and comments on their ability to handle noise: 'The generative model does not model additive noise; we assume that the noise is explicitly modeled as a latent variable and its effect is manifested through [the ground truth generator], as is done by [several citations].” I admit that this might not necessarily be a contradictory claim, but I nonetheless would appreciate it if the authors clarified how their method of explicitly modeling additive noise is beneficial over the type of setup in {1} and the citations in Locatello et al.’s"**. First, with *'all models cited above'*, we were referring only to the nonlinear ICA models cited directly above . We will clarify this in final version. We will add citation for Locatello and explain how it differ from ours. In partcular: as for our handling of noise, we wish to emphasize that we are explicitly modelling observational noise, that is, additive noise which is added to the measurements due to imperfections in measurement devices and is ubiquitous in many practical applications such as MEG data. This is in stark contrast to the more loose meaning of noise, as in Locatello, where noise means some general stochasticity that is treated as another latent variable -- their approach would thus be ill-suited to the type of denoising one would often need in practice.
>
> lines 52-53: In order to make our contribution clearer we will better explain it with the following sentence: "Furthermore, the framework guarantees identifiability of a rich class of nonlinear ICA models that is able to exploit dependency structures of any arbitrary order and thus, for instance, extends to spatially structured data."
>
> line 67: We will change our writing to stress that we are referring to nonlinear ICA only by changing to: "It achieves the following very practical properties which have previously been unattainable in the context of nonlinear ICA: the ability to account for both nonstationarity and autocorrelation in a fully unsupervised setting...[rest unchanged]"
>
> lines 96-99: *"1. b) Notice that the mixing function is assumed bijective and thus dimension reduction is not possible in most of the above models. The only exception is Khemakhem et al. who achieve this by assuming that we know the distribution of some additive noise on the observation".* **"...I find the second part of this quote (“dimension reduction is not possible”) surprising in general, considering that any publication attempting to perform disentanglement on the DisentanglementLib dataset necessarily has to assume dimensionality reduction. I thought maybe the authors intended to say that the theory does not allow dimensionality reduction, even if the practical implementations do. However, Klindt et al., which is cited earlier, presents an identifiable disentanglement framework that only assumes an injective mixing function in their theory and therefore allows for dimensionality reduction."**. We were indeed talking about identifiable dimension reduction in nonlinear ICA: our theoretical identifiability results explicitly allow dimension reduction (the mixing function takes value in a manifold which can have lower dimension than the ambient dimension), which is not the case for vast majority of disentanglement works as they have unconditional factorial priors and thus by definition unidentifiable -- we will add this explanation. Khemakhem and Klindt are exceptions here; we will add citation and further comments on Klindt paper into this paragraph since, as you pointed out, it is currently not mentioned with respect to dimension reduction.
>
> line 110: We will cut out the second half of this sentence as it is explained in more detail elsewhere in the paper.
>
> lines 123-125: *'under the conditions given in the next section, we can now guarantee identifiability for a very broad and rich class of models. First, notice that all previous Nonlinear ICA time-series models can be recast and often improved upon when viewed through this new unifying framework.'* **"1.c) I do not understand how this could possibly be true given that not all previous Nonlinear ICA work abides by the conditions given in sections 3 and 4. Just as two examples, the requirement for unconditional independence on lines 114-115, and tail behavior in assumption A1 are not ubiquitous in the literature. I could definitely be misunderstanding what the authors intended with their statement, in which case I would be happy with a clarification."**. Here we provide clarification: we are not saying that the previous models can be restated in their exact form in our framework; by *'recast'* we meant that we can create models that are very much like those previous works, and capture their dependency profiles, except with the changes that by assuming unconditional independence and output noise we now allow them to perform dimension reduction. The tradeoff of this is having to make the additional assumptions of our theorem. We will alter our writing to clarify this.
>
> line 139, and lines 141-142: In order to better support and clarify our statement, we will combine the above into a more measured sentence: "Next we propose one which combines the following properties of previous nonlinear ICA models into a single model: ability to account for both nonstationarity and autocorrelation in a fully unsupervised setting, to performs dimensionality reduction and models hidden states"
>
> line 166: We will remove this sentence as the same thing is explained in more detail at the end of the Section 4., and in Section 6.
>
> lines 339-341: This is under the "Related Works" section and we thus believe it is important here to explain to the reader what are our significant, novel, contributions. We agree that this is too vague, however, and will instead rephrase it as follows: "Previous works on nonlinear ICA have exploited autocorrelations [citations] and nonstationarities [citations] for identifiability. The SNICA setting provides a unifying framework which allows for both types of temporal dependencies, and further, extends identifiability to any arbitrary higher order data structures which has not previously been considered in the context of nonlinear ICA.".
>
> 2. Comments on experiments
>
> **"This leaves me wondering why they felt that this was a fair comparison, given that the problem setup is not at all matched to what IIA-HMM was designed to solve. I would be curious to see how they do for a non-dimensionality-reduced setup...."**. The first aim of these simulated experiments is to show that our model functions as expected on the back of our theory, while the real data experiment provides a much fairer comparison to IIA-HMM. In fact, *any* other model would be naturally disadvantaged on data simulated from SNICA -- we decided to use IIA-HMM as it is the only other *identifiable* model that can handle similar temporal dependencies. In order to show that the results are not only due to PCA-preprocessing, we have now run additional experiment where there is no dimension reduction and will include this in our final version. SNICA still outperforms IIA-HHM also in this setting, although the latter's performance is now improved for small dimensionality (3d: e.g.  MCC avg. 0.4 for 3 mixing layers), though remains clearly below SNICA (3d: MCC avg. 0.7). IIA-HMM performance for dimensions above 6 is very poor (MCC < 0.3) suggesting it's poor performance is not solely due to PCA. These results will be added in the final version.
>
> **"Or, a better solution would be for them to compare their model against an alternative that is appropriately matched to the dimensionality reduction task."** To address this, we have also decided to include iVAE as an additional benchmark. iVAE is able to perform dimension reduction but the reason we did not include it initially is that it cannot handle latent variables, nor their temporal dependencies. To resolve this, we have allowed iVAE to "cheat" by supplying the model with the latent HMM states as its observed auxiliary variables. This gives us a very challenging benchmark. We are currently in the process of finalizing these simulations and will inform you of the results as soon as they are completed.
>
> **"Furthermore, the authors do not provide a baseline comparison for their denoising task. This might be because the authors wished to focus on identifiable models, which restricts the otherwise large set of denoising methods..."** Indeed, we wished the focus on identifiable models. As explained above, we have now however added iVAE as a challenging benchmark which will be added as a benchmark here. Since iVAE is still likely to suffer from its limited ability to handle temporal dependencies, we have also included standard Kalman filter as a naive, and well-known, baseline for the denoising task. Together these two models give good benchmarks on this task. Finally, our aim is not in developing state-of-the-art denoising model but rather show that it is one of the capabilities provided by a fully probabilistic framework.
>
> **"Minor complaints"**: We acknowledge these fully and will address all of them.

---

> > ### Comment · Reviewer_yFEi · 2021-08-19
> > **Concerns met -- score adjusted.**
> >
> > I appreciate the careful response to my review. The proposed sentence revisions address all of my concerns regarding the claimed scope of contribution. The additional experiments proposed will allow for a much more well-rounded understanding of their contributed performance. Thank you, and I look forward to seeing the iVAE & Kalman comparisons in the final version. I have raised my score by two points.

---

### Official Review · Reviewer_8dTg · 2021-07-16

**Rating:** 7
**Confidence:** 3

**Summary:**

In this work, the authors present an identifiability framework called Structured Nonlinear Independent Component Analysis (SNICA), which extends previous identifiability theory for time-series models into a much broader class of models.

**Ethical Concerns:**

None.

**Limitations And Societal Impact:**

Yes.

**Main Review:**

The manuscript is very well written and structured, and is accessible to readers even if they cannot follow all the technical details. The work aims to contribute to the growing area of identifiability models, a field that is starting to gain traction as a basis for a more rigorous understanding of representation learning with nonlinear parametrised models (such as neural networks).

One confusion I had when reading the manuscript was a seeming contradiction between the definition of structured nonlinear ICA and the nonstationary models presented afterwards. In particular, line 114 assumes that the latent components are independently distributed over time, i.e. knowing the latents at time t yield no information about the latents at time t + 1. Yet this assumption is then being made in equation (4), and is crucial to cover previous identifiable models such as TCL. I'd appreciate if the authors could clarify this point. I am happy to raise my score if this point can be clarified.

Second, it would be good to give some more intuition on how the two-step procedure described in section 4 (first remove noise, than identify latents) is converted into a practical algorithm. Clearly, for each individual observable x_t one cannot remove the noise, Theorem 1 just shows that the distribution can be uncovered. Is this a step that is explicitly done during model training, or is implicit?

One crucial aspect currently missing in the manuscript is a clear exposition for how the training/estimation actually works. There is only a brief paragraph at the beginning of section 5 stating that "by framing the model within conjugate exponential families we are able to perform learning and inference using Structured VAEs". Currently, this reviewer feels there is no way a reader would be able to replicate the experiments without the code itself. While I appreciate that the authors promised to release the code, I feel the manuscript should contain a clear exposition for the complete training protocol such that replication would be possible without looking at the code.

## Rebuttal
The rebuttal clarified my concerns and questions. I am raising my score by one point.

**Time Spent Reviewing:**

2

---

> ### Author Response · Authors · 2021-08-10
> **Author response #1**
>
> Dear reviewer, we would like to thank you for these excellent comments and questions. Please see below our responses:
>
> **"In particular, line 114 assumes that the latent components are independently distributed over time, i.e. knowing the latents at time t yield no information about the latents at time t + 1. Yet this assumption is then being made in equation (4), and is crucial to cover previous identifiable models such as TCL."**. We would like to clarify that line 114 does *not* assume independent distribution over time; what is meant by this formula is that the N latent processes  $(s_t^{(i)})_{t \in \mathbb{T}}$ (for $i=1,\dots,N$) are independent over $i$, but for any given $i$ and $t, u \in \mathbb{T}$, there is no reason that $s_t^{(i)}$ and $s_u^{(i)}$ would be independent. Let us also point out that the indexing set $\mathbb{T}$ is not restricted to $\mathbb{T} = \mathbb{N}$, as is the case for time-series, but could for instance be the set of vertices of a graph (e.g. Fig. 1c.).
>
> **"Second, it would be good to give some more intuition on how the two-step procedure described in section 4 (first remove noise, than identify latents) is converted into a practical algorithm. Clearly, for each individual observable x\_t one cannot remove the noise, Theorem 1 just shows that the distribution can be uncovered. Is this a step that is explicitly done during model training, or is implicit?"**. This is done implicitly in the sense that the two steps (denoising and demixing) are done concurrently, which is one of the benefits of framing our model in fully probabilistic manner. Our learning framework integrates inference of noise-free components as part of the learning of the mixing and other parameters as follows: the framework computes the posterior distribution of the independent components given the observed data $p(\mathbf{s} | \mathbf{x})$. This we get by learning a decoder network of the Structured VAE (SVAE), which maps from the observed data to the natural parameters of the independent components' distribution. (Once we have learned the decoder, we can use it to obtain posterior of the denoised mixed data as well, if necessary, and we can also take any new data and denoise it. If desirable, one could then extract the mean or the median of the posterior for point estimates.). See Appendix C, for more detail. If our paper is accepted, we will include more of this detail in the main part as well.
>
> **"One crucial aspect currently missing in the manuscript is a clear exposition for how the training/estimation actually works. There is only a brief paragraph at the beginning of section 5 stating that 'by framing the model within conjugate exponential families we are able to perform learning and inference using Structured VAEs'. Currently, this reviewer feels there is no way a reader would be able to replicate the experiments without the code itself. While I appreciate that the authors promised to release the code, I feel the manuscript should contain a clear exposition for the complete training protocol such that replication would be possible without looking at the code."** We have presently left our training details in the Appendices C and D simply due to space limitations and the fact that Structured VAE is an already published algorithm. We do appreciate your point however, especially given that ours is a special application of it. Should our paper be accepted, we would utilize the extra page to include in the main paper the central parts of Appendix C along with explanation and in particular the training details form Appendix D. Also, if there is a way to anonymously  share code with you (non-github) then we would be glad to do so during this review period if this would be helpful.

---

> > ### Comment · Reviewer_8dTg · 2021-08-13
> > **Thanks!**
> >
> > Thanks a lot for the clarifications! I am raising my score by one point.

---

### Official Review · Reviewer_9Cro · 2021-07-18

**Rating:** 8
**Confidence:** 4

**Summary:**

The authors provide an identifiability result for non-linear ICA for a class of time series models. There is a clear contribution with respect to previous work that was typically restricted to condionnally independent sources, conditionned on a hidden markov chain in finite stare spaces. This new result allows identifying independent sources with a specific form of time dependency.

**Limitations And Societal Impact:**

Limitations are adequately addressed.

**Main Review:**

The result seem to build on a recent identifiability result by Gassiat et al. for Markov chains in general state space, which one could have expected to open the way to new ICA identifiability, in line with what has happened in the past for HMMs with finite state space. However, the authors also provide a slightly more general result in theorem 1 (allowing supergaussian source distributions) before exploiting it for ICA identifiability. The result is restricted to a specific source model (Markov switching linear model) but one could argue that it is a good step forward for addressing more general cases.

**Time Spent Reviewing:**

3 hours

---

> ### Author Response · Authors · 2021-08-10
> **Author response #1**
>
> Dear reviewer, we very much appreciate your positive comments. Our only addition is to the following point you make: **"The result is restricted to a specific source model (Markov switching linear model) but one could argue that it is a good step forward for addressing more general cases."**. With reference to this, we would like to highlight that while our delta-SNICA model indeed includes Markov switching, our identifiability theorems and the general SNICA framework are not restricted to Markov switching models but are much more general; see for instance the model in Fig. 1b.) which uses stationary dependencies (such as those in autoregressive models). Further, our work is not restricted to time-series -- the SNICA identifiability framework's novelty is that it allows any structured data as long as there is appropriate index over which the data has dependencies (details in Sec. 3.1). This could, for instance, be two dimensional spatial data (e.g. spatial dependencies in image data) or some other process on an appropriate graph (Fig. 1c.). In summary, our contribution is to establish very general conditions under which nonlinear ICA is identifiable for structured data.

---

> > ### Comment · Reviewer_9Cro · 2021-08-27
> > **Good point**
> >
> > My apologies for the unfaithful summary of the scope of the results. Indeed the identifiability theorems are more general, and I did not mean to exclude autoregressive models when I mentioned Markov switching models. As you state in the paper "While the above framework has great generality, any practical application will need a specific model" and you investigate a specific one with $\Delta$-SNICA and also show how the novel general SNICA framework encompasses previous identifiability results. After reconsidering the generality of the results, I raise my evaluation to 8. Assumptions are quite technical but will likely help showing identifiability of (spatially-, time- or else) structured models of practical relevance in the future.

---

### Decision · Program_Chairs · 2021-09-27

**Decision:**

Accept (Poster)

**Comment:**

The paper extends the literature on identifiable nonlinear ICA for time series in ways that all reviewers found satisfactory. There is a clear consensus that the work is novel and interesting, and many of the reviewers were convinced by the authors' rebuttal to increase their score even further. Therefore I recommend acceptance.